# Multi-Dimensional Insights: Benchmarking Real-World Personalization in Large Multimodal Models

## Abstract

The rapidly developing field of large multimodal models (LMMs) has led to the emergence of diverse models with remarkable capabilities. However, existing benchmarks fail to comprehensively, objectively and accurately evaluate whether LMMs align with the diverse needs of humans in real-world scenarios. To bridge this gap, we propose the **M**ulti-**D**imensional **I**nsights (MDI) benchmark, which includes over 500 images covering six common scenarios of human life. Notably, the MDI-Benchmark offers two significant advantages over existing evaluations: (1) Each image is accompanied by two types of questions: simple questions to assess the model's understanding of the image, and complex questions to evaluate the model's ability to analyze and reason beyond basic content. (2) Recognizing that people of different age groups have varying needs and perspectives when faced with the same scenario, our benchmark stratifies questions into three age categories: young people, middle-aged people, and older people. This design allows for a detailed assessment of LMMs' capabilities in meeting the preferences and needs of different age groups. With MDI-Benchmark, the strong model like GPT-4o achieve 79% accuracy on age-related tasks, indicating that existing LMMs still have considerable room for improvement in addressing real-world applications. Looking ahead, we anticipate that the MDI-Benchmark will open new pathways for aligning real-world personalization in LMMs.

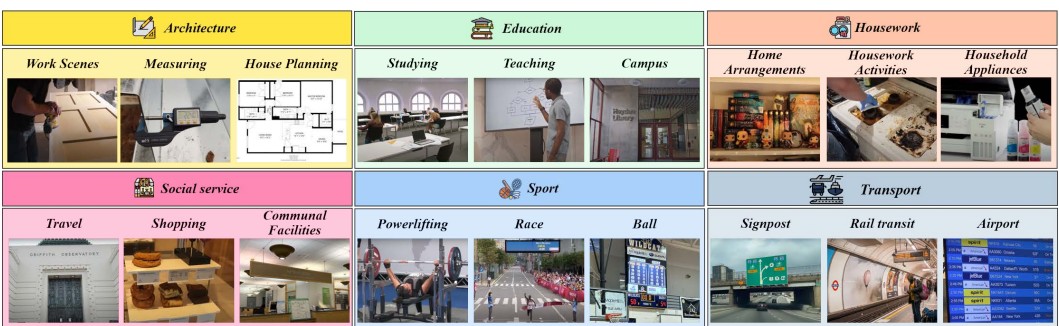

Figure 1: The overview of the MDI Benchmark's six real-world multimodal scenarios, each comprising three sub-domains.

## 1 Introduction

Developing personalized artificial intelligence (AI) assistants to address the diverse needs of different users has long been a significant pursuit for humanity (Kobsa & Schreck, 2003; Xiao et al., 2018; Kocaballi et al., 2019; Rafieian & Yoganarasimhan, 2023; Pesovski et al., 2024). In real-world scenarios, an ideal AI assistant should be capable of precisely meeting the specific demands of individuals across various age groups, cultural backgrounds, and professional fields.

Recently, the field of artificial intelligence has undergone a significant paradigm shift, transitioning from specialized small models designed for specific simple tasks (Rawat & Wang, 2017; Zhao et al., 2019; Minaee et al., 2021; Singh et al., 2017) to unified large multimodal models (LMMs) capable of handling complex tasks (Zhang et al., 2024). This paradigm shift marks a crucial step toward achieving Artificial General Intelligence (AGI) and underscores the potential for LMMs to become personalized human assistants.

To comprehensively evaluate the capabilities of LMMs, researchers have constructed several common visual question-answering benchmarks (Goyal et al., 2017; Chen et al., 2015; Marino et al., 2019; Mishra et al., 2019; Biten et al., 2019) that assess general image-text comprehension and dialogue capabilities of LMMs. However, these benchmarks merely compare answers to standard solutions, offering limited insights into the fine-grained capabilities of models. To address this limitation, subsequent multimodal understanding benchmarks are developed (Yu et al., 2023; Liu et al., 2023; Fu et al., 2024a; Ying et al., 2024), covering a broader range of tasks and a larger number of test samples. This refinement enables a more precise evaluation of model capabilities, fostering the development of more robust LMMs. Nevertheless, current benchmarks focus primarily on technical metrics for specific tasks, neglecting two critical research questions:

***Q1:*** *Can these LMMs truly align with the actual needs of humans in real-world scenarios?*

***Q2:*** *Can these LMMs subsequently address the diverse needs of distinct groups?*

To tackle these challenges, we introduce a novel "**M**ulti-**D**imensional **I**nsights" (MDI) benchmark, which encompasses various real-world scenarios, different problem complexities, and diverse age groups. In detail, the MDI-Benchmark consists of more than 500 real-world images and 1.2k human-posed questions. As shown in Figure 1, it covers six major scenarios of human life: Architecture, Education, Housework, Social Services, Sport, and Transport. Furthermore, MDI-Benchmark focuses on evaluating LMMs from the following two dimensions:

**Question Complexity Dimension.** This dimension categorizes human-posed problems into two levels of complexity. The first level assesses the basic capabilities of LMMs, such as object detection and optical character recognition (OCR), etc. The second level evaluates more complex capabilities, including logical reasoning, mathematical calculation, and knowledge application.

**Age Dimension.** Age is a fundamental criterion for evaluating individual differences, as people of different ages have diverse needs. We categorize individuals into three age groups: young people, middle-aged people, and older people, to assess the effectiveness of LMMs in addressing the varying needs and preferences across these groups. Our goal is to comprehensively assess whether LMMs can meet the diverse needs of humans in practical situations.

In summary, our major contributions are listed:

- To align with the actual needs of humans for Large Multimodal Models, we are the first to propose a multi-modal benchmark for providing a thorough assessment of the capacities of LMMs in practical, real-world scenarios.

- The MDI-Benchmark includes over 500 real-world images and 1.2k human-posed questions, spanning six real-world multimodal scenarios. Each scenario is divided into 3 sub-domains with 2 levels of complexity. Additionally, we incorporate age factors into the evaluation to guide LMMs in personalizing their responses for different demographic groups.

- With the MDI-Benchmark, we conduct a comprehensive evaluation of several mainstream LMMs. Specifically, GPT-4o achieved the best results across all indicators, but there is still significant room for improvement in addressing the needs of different age groups. Further analysis across dimensions such as *Scenario*, *Complexity* and *Age* provides valuable insights for developing reliable, personalized human assistants.

We hope our research will advance the application of multimodal large models in real-world scenarios and pave the way for the development of multi-dimensional personalization.

## 2 RELATED WORK

### 2.1 MULTIMODAL DATASET AND BENCHMARK

To evaluate the capabilities of LMMs, a variety of benchmarks from past research have been applied. Among them, Flickr30k (Young et al., 2014), COCO Captions (Chen et al., 2015), and Nocaps (Agrawal et al., 2019) are utilized to evaluate LMMs' text generation and image description abilities. Vizwiz (Bigham et al., 2010), VQA (Goyal et al., 2017), GQA (Hudson & Manning, 2019), and OK-VQA (Marino et al., 2019) are used to assess LMMs' comprehension of image information and question-answering abilities. For evaluating OCR capabilities, benchmarks like ST-VQA (Biten et al., 2019) and OCR-VQA (Mishra et al., 2019) are employed. DocVQA (Mathew et al., 2021) is specifically used to evaluate a model's ability to understand and identify documents.

To further explore the fine-grained capabilities of LMMs, recent benchmarks have significantly expanded the types of tasks assessed. Examples of such benchmarks include LVLM-eHub (Xu et al., 2023), MM-Vet (Yu et al., 2023), MMBench (Liu et al., 2023), SEED-Bench (Li et al., 2023), MME (Fu et al., 2024a), MMT-Bench (Ying et al., 2024), Video-MME (Fu et al., 2024b), MMMU (Yue et al., 2023), MMMU-Pro (Yue et al., 2024), MathVista (Lu et al., 2024b), Contextual (Wadhawan et al., 2024), We-Math(Qiao et al., 2024), and MMEvol(Luo et al., 2024). Nevertheless, it should be noted that these benchmarks have not fully explored the capability of LMMs to address the diverse needs of different individuals. Therefore, we hope to better explore this ability through the MDI-Benchmark.

### 2.2 LARGE MULTIMODAL MODELS

Building on the success of many large language models (LLMs) (Brown et al., 2020; Touvron et al., 2023; Chiang et al., 2023), recent research has combined large language models with visual encoders to form LMMs with powerful visual understanding and semantic generation capabilities. Many excellent open-source (Hong et al., 2023; Wang et al., 2023; Hu et al., 2024; Lu et al., 2024a; Liu et al., 2024b; Ye et al., 2023; Abdin et al., 2024) and closed-source (Team et al., 2023; Bai et al., 2023; OpenAI, 2023; 2024) projects have been developed. This development has further enhanced the potential for realizing personalized AI assistants.

### 2.3 PERSONALIZED RESEARCH

To achieve personalized AI assistants, large language models (LLMs) are currently attempting to combine with users' personalized outputs to enhance their personalization capabilities and enable them to generate outputs that conform to users' preferences (Woźniak et al., 2024; Zhuang et al., 2024; Baek et al., 2024; Tan et al., 2024). Simultaneously, to further expand the understanding ability of LLMs in the face of different needs, personalized data generation is also crucial(Chan et al., 2024). In this work, we utilize the MDI-Benchmark to evaluate the ability of existing large multimodal models to address personalized needs and provide our insights for future LMMs research.

## 3 MDI-BENCHMARK

The benchmark sample design emphasizes the real-world complexity of information, scene variability, and age differences. People's information concerns often vary by scenario. As shown in Figure 2, a family buying a new house may focus on practical issues that are closely related to them, such as kitchen type, garage capacity, and bedroom amenities. Spectators at sports events may concern themselves with game details, player achievements, and game progress.

### 3.1 EVALUATION DIMENSION

In contrast to existing work, MDI-Benchmark emphasizes the model's performance on real-world problems across various ages and complexities within specific task scenarios, it is structured along three different dimensions: scenario, age, and problem complexity.

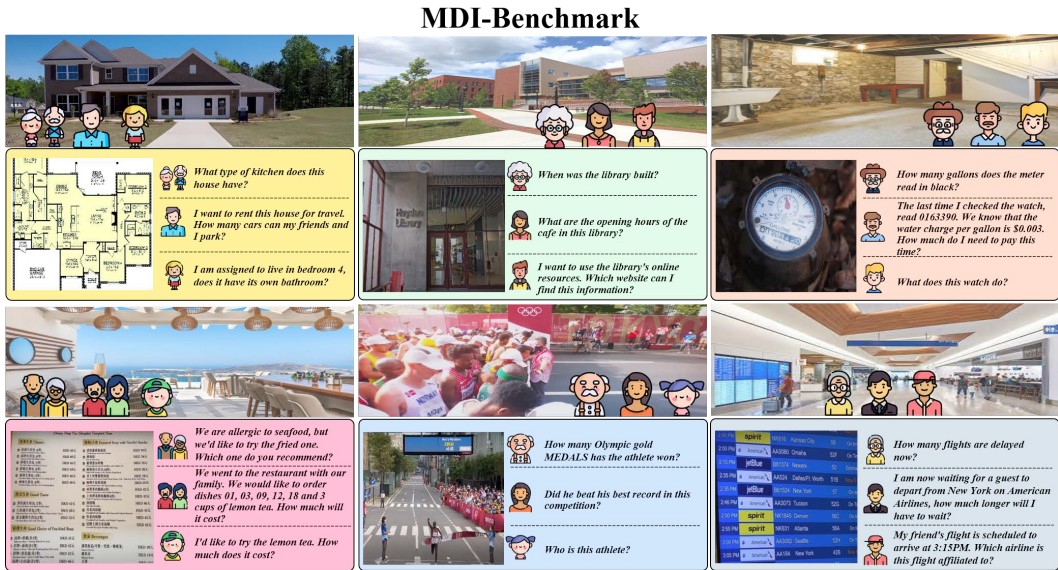

Figure 2: The MDI-Benchmark includes real needs of different age groups in six major real-world scenarios.

**Scenario Dimension.** From the perspective of the scenario, the MDI-Benchmark aims to closely align with the real needs of human life. Unlike the capability evaluation focus of previous LMMs evaluation benchmarks, the MDI-Benchmark is constructed based on real-life scenarios.

In response to the various scenarios that humans face in real life, we have drawn on the definitions provided in sociological literature (Tajfel, 1979; Birmingham et al., 2008; Spears, 2021) and expanded upon them to identify 30 sub-domain scenarios. On this basis, we conducted a one-month questionnaire survey covering people of different ages, genders, and occupations. A total of 2,500 questionnaires were distributed, and 2,374 valid responses were collected. Based on the frequency of sub-domain selection in the questionnaires, we selected the top 18 sub-domains, which were ultimately summarized into six main scenarios: architecture, education, housework, social service, sports, and transport. We collected images from these subdomains to ensure this benchmark is rich in scenario information. Examples are in the Appendix B.1.

**Problem Complexity Dimension.** In the realm of everyday human activities, the level of complexity varies significantly, and the definition of difficulty is often subjective. To streamline this definition, we have quantified the problems hierarchically based on the fundamental capabilities of the model as the atomic units. Based on this criterion, we have filtered survey questions and refined previous evaluation standards. Furthermore, the MDI-Benchmark is categorized into two levels: (1) The first level involves relatively straightforward problem types that mainly evaluate the model's ability to extract scenario information. This includes tasks such as detection, optical character recognition, position recognition, color recognition, and other fundamental capacities. (2) The second level demands that the model skillfully analyze both scenario information and user semantic information with logical acuity while integrating relevant knowledge to effectively meet user requirements. Examples are in the Appendix B.2.

**Age Dimension.** Age is a universal and specific criterion for group classification, making it more objective compared to classifications based on culture and religious beliefs. As a fundamental attribute possessed by everyone, age is easy to quantify and compare. By using age as a classification dimension, we can better understand the needs of various groups and assess the capability of LMMs to meet these diverse needs. For the purposes of assessment and quantification, we identified three distinct age groups: young people (ages 10-25), middle-aged people (ages 35-50), and old people (ages 60-75). We engaged individuals from these age brackets in real-life scenarios to inquire about their needs. The results of these surveys informed the creation of the initial version of the MDI-Benchmark. Examples are in the Appendix B.3.

## 3.2 DATA COLLECTION

**Data Source.** Existing LMMs evaluation benchmarks have been widely used to evaluate and train new models. To ensure the accuracy of the evaluation results, we collected over 500 new images that were not included in existing datasets and recruited 120 volunteers from three age groups. From each group, we sampled 10 volunteers to form a 30-person data construction team. The main data collection process was as follows: First, after determining the scenario dimension information, the data construction team wrote detailed scenario information based on their interests. Meanwhile, we input the scenario dimension information into open-source models (e.g., GPT-4o, Gemini 1.5 Pro) and closed-source models (e.g., LLaVA-Next, MiniCPM) to generate more personalized, diverse, and detailed scenario descriptions. Furthermore, the descriptions created by both humans and models were used as keywords to search for relevant images on the Internet. Meanwhile, We paid volunteers a sufficient wage, approximately seven dollars per hour. These volunteers were tasked with categorizing the images into six scenario dimensions. To ensure data balance and minimize bias, we ensured diversity within each age group in terms of gender, occupation, and other factors. Detailed classification standards and guidelines were provided to ensure consistency in categorization. We employed a cross-validation approach, whereby each group of volunteers screened the images, and we retained only those images that were categorized identically by all three groups. Additionally, multiple iterations of validation were conducted. This comprehensive process helped to construct a balanced and reliable data source.

**Question and Answer Generation.** After obtaining the collected images, we used a heuristic method to manually generate questions and problems. The specific process is as follows: (1) Construction of Knowledge Base. Specifically, multiple open-source and closed-source models are first used to describe the scenario content in the image and are summarized by human experts. Subsequently, additional information related to the scenario content was found through an Internet search, and the image and this information were combined to form a knowledge base. (2) Generation of Difficult Multi-Choice Questions. To ensure the consistency of the generated questions with the image content, we invited volunteers from three different age groups who participated in the data collection phase to submit questions. These volunteers posed questions of varying complexity based on the image scenarios and knowledge base content and created confusing incorrect options. (3) Question Format. The image-question pairs provided by the volunteers had to follow the format: [Level]-[Age]-[Scenario]. Here, Level includes level 1 and level 2; Age includes old, mid, and young; Scenario includes architecture, education, housework, social services, sports, and transport. Finally, a team of experts screened and evaluated the questions submitted by the volunteers to finalize the construction of the questions.

**Data Statistics.** The MDI-Benchmark is collected from three different dimensions: scenarios, age groups, and abilities. It includes a total of 514 images and 1298 questions, all newly collected. Meanwhile, we strived to ensure a balance of data across different scenarios, ages, and question complexities. The detailed information is presented in the Table 1. As shown in Figure 2, the dataset covers six domains, each with three sub-domains, providing a comprehensive and structured construction of data across various fields.

Table 1: Statistical details of MDI-Benchmark.

| Scenarios | Number of images | Number of L1 questions | Number of L2 questions | Number of old questions | Number of mid questions | Number of young questions |
|---|---|---|---|---|---|---|
| Architecture | 85 | 121 | 112 | 77 | 74 | 82 |
| Education | 85 | 114 | 115 | 80 | 79 | 70 |
| Housework | 86 | 103 | 109 | 71 | 74 | 67 |
| Social services | 86 | 95 | 108 | 65 | 66 | 72 |
| Sports | 86 | 107 | 103 | 70 | 73 | 67 |
| Transport | 86 | 109 | 102 | 73 | 70 | 68 |
| Total | 86 | 649 | 649 | 436 | 436 | 426 |

## 4 EXPERIMENTS

### 4.1 EXPERIMENTAL SETTINGS

**Evaluation Protocols.** To effectively evaluate the model's output, we require the model to provide the correct answer in its response. The specific prompt information is shown in the Table 3. Based

Table 2: LMMs Performance on MDI-Benchmark in Terms of Level and Scenario. Vertically, the table is composed of a model score and two Level sub-tables, where the model score is obtained from Formula 1. Each sub-table consists of seven columns showing the accuracy rates of LMMs in different scenarios. The first column of each sub-table represents the mean value of the subsequent six columns, reflecting the overall performance at different levels. The annotations for Level and Scenario are as follows: Level 1: assessment questions that focus only on basic perceptual ability; Level 2: assessment questions that involve logical reasoning. The scenarios are abbreviated as follows: Arc (architecture), Edu (education), Hou (housework), Soc (social service), Spo (sport), Tra (transport). Horizontally, the table is divided into two blocks. For better statistics and analysis, we will display the blocks as closed-source model statistics and open-source model statistics. The best performance in each block is highlighted in blue and green.

| Model | Final Score | Level 1 | | | | | | | Level 2 | | | | | | |
|---|---|---|---|---|---|---|---|---|---|---|---|---|---|---|---|
| | | Avg | Arc | Edu | Hou | Soc | Spo | Tra | Avg | Arc | Edu | Hou | Soc | Spo | Tra |
| *Closed-source* | | | | | | | | | | | | | | | |
| **GPT-4o** | 78.46 | 87.46 | 76.47 | 94.12 | 92.16 | 90.20 | 86.27 | 94.12 | 69.45 | 70.59 | 70.59 | 78.43 | 82.35 | 54.90 | 66.67 |
| **GPT-4V** | 74.92 | 87.46 | 86.27 | 92.16 | 86.27 | 90.20 | 88.24 | 90.20 | 62.38 | 72.55 | 70.59 | 74.51 | 60.78 | 45.10 | 56.86 |
| **Gemini 1.5 Pro** | 69.13 | 82.32 | 68.63 | 92.16 | 76.47 | 88.24 | 86.27 | 90.20 | 55.95 | 52.94 | 56.86 | 54.90 | 74.51 | 43.14 | 58.82 |
| **Qwen-VL-Plus** | 43.57 | 56.59 | 43.14 | 64.71 | 62.75 | 78.43 | 50.98 | 45.10 | 30.55 | 35.29 | 41.18 | 37.25 | 25.49 | 23.53 | 23.53 |
| *Open-source* | | | | | | | | | | | | | | | |
| **LLaVA-NeXT-110B** | 65.59 | 79.10 | 60.78 | 92.16 | 78.43 | 84.31 | 78.43 | 88.24 | 52.09 | 66.67 | 56.86 | 54.90 | 64.71 | 31.37 | 43.14 |
| **LLaVA-NeXT-72B** | 63.67 | 76.21 | 68.63 | 88.24 | 80.39 | 82.35 | 70.59 | 74.51 | 51.13 | 66.67 | 54.90 | 52.94 | 60.78 | 33.33 | 43.14 |
| **MiniCPM-LLaMA3-V 2.5** | 55.95 | 72.67 | 52.94 | 86.27 | 70.59 | 82.35 | 70.59 | 80.39 | 39.23 | 45.10 | 49.02 | 49.02 | 31.37 | 27.45 | 37.25 |
| **mPLUG-Owl2-7B** | 52.57 | 64.63 | 49.02 | 70.59 | 74.51 | 70.59 | 58.82 | 70.59 | 40.51 | 41.18 | 41.18 | 47.06 | 39.22 | 29.41 | 49.02 |
| **DeepSeek-VL-7B** | 52.09 | 68.49 | 49.02 | 70.59 | 74.51 | 80.39 | 62.75 | 80.39 | 35.69 | 41.18 | 33.33 | 39.22 | 41.18 | 21.57 | 41.18 |
| **Phi3-Vision-4.2B** | 50.80 | 67.20 | 50.98 | 76.47 | 60.78 | 80.39 | 62.75 | 78.43 | 34.41 | 37.25 | 33.33 | 41.18 | 43.14 | 21.57 | 33.33 |
| **CogVLM-chat** | 49.84 | 60.77 | 49.02 | 72.55 | 62.75 | 56.86 | 68.63 | 60.78 | 38.91 | 49.02 | 33.33 | 43.14 | 41.18 | 27.45 | 43.14 |
| **DeepSeek-VL-1.3B** | 46.30 | 58.20 | 45.10 | 56.86 | 66.67 | 56.86 | 66.67 | 62.75 | 34.41 | 35.29 | 29.41 | 29.41 | 39.22 | 27.45 | 49.02 |
| **CogAgent-vqa** | 41.16 | 49.52 | 35.29 | 45.10 | 66.67 | 54.90 | 56.86 | 43.14 | 32.80 | 31.37 | 35.29 | 35.29 | 37.25 | 25.49 | 35.29 |
| **LLaVA-NeXT-7B** | 33.60 | 43.09 | 31.37 | 52.94 | 43.14 | 49.02 | 39.22 | 47.06 | 24.12 | 35.29 | 13.73 | 37.25 | 23.53 | 9.80 | 27.45 |

Table 3: Prompt templates for response generations.

| Type | Prompt Template |
|---|---|
| Multiple Choice | Now, we require you to solve a multiple-choice real-world question. Please briefly describe your thought process and provide the final answer(option). **Question**: <Question> **Option**: <Option> Regarding the format, please answer following the template below, and be sure to include two <> symbols: **<Thought process>**: <<your thought process>> **<Answer>**: <<your option>> |

on this, the accuracy of the response was calculated. This means that if the model articulates the correct concept but fails to produce the precise answer, it will be classified as incorrect. This approach underscores the model's ability to follow instructions accurately, highlighting any deficiencies in this capacity. In addition, since the prompt input format varies across different models, we investigated the input format for each model. We then endeavored to maintain consistency in the prompts, adhering to the official input format provided by each model. This approach aims to minimize the impact of prompt differences on model performance.

**Prompt Template.** Table 3 report the prompt templates in our experiments.

**Evaluation Models.** We studied the performance of two different categories of base models on the MDI-Benchmark. (a) Closed-source models: GPT-4o(OpenAI, 2024), GPT-4V(OpenAI, 2023), Qwen-VL-Plus(Bai et al., 2023), Gemini 1.5 Pro(Team et al., 2023) (b) Open-source models: LLaVA-NeXT-110B(Liu et al., 2024a), LLaVA-NeXT-70B(Liu et al., 2024a), LLaVA-NeXT-7B(Liu et al., 2024b), DeepSeek-VL-7B, DeepSeek-VL-1.3B(Lu et al., 2024a), Phi3-Vision-4.2B(Abdin et al., 2024), MiniCPM-LLaMA3-V 2.5(Hu et al., 2024), CogVLM-chat(Wang et al., 2023), CogAgent-vqa(Hong et al., 2023), mPLUG-Owl2-7B(Ye et al., 2023)

**Scoring Metric.** Table 2 shows the overall performance of different LMMs under two levels of problem complexity and across six scenarios. To better assess the capabilities demonstrated by the model, we defined the scoring metric:

$$\text{Score}_{\text{final}} = \alpha \cdot \text{Score}_{\text{L1}} + (1 - \alpha) \cdot \text{Score}_{\text{L2}} \tag{1}$$

where $\text{Score}_{\text{L1}}$, $\text{Score}_{\text{L2}}$ denotes the average performance of LMMs in various fields at the first and second tiers, respectively and we set the default value of $\alpha$ to 0.5.

## 4.2 MAIN RESULTS

Table 2 illustrates the overall performance of different LMMs on MDI-benchmark. We find out the following insights:

**GPT family demonstrate an absolute advantage.** GPT-4o leads all models and receives the highest performance score. It can also be observed that closed-source models generally outperform open-source models. However, some powerful open-source models are struggling to catch up with closed-source models. For example, the LLaVA-NeXT-110B, and LLaVA-NeXT-72B performed slightly worse than the Gemini 1.5 Pro and better than the Qwen-VL-Plus.

**Scaling phenomenon of model performance.** Furthermore, due to the limited data available for the closed-source models, we observed some interesting trends among the open-source models. We selected the best-performing open-source models in various sizes, from LLaVA-NeXT-110B and LLaVA-NeXT-72B to MiniCPM-LLaMA3-V 2.5, DeepSeek-VL-7B, Phi3-Vision-4.2B and DeepSeek-VL-1.3B. As shown in Figure 4 (the Leaderboard of different LMMs), the final scores for these models showed that the larger the model parameters, the better its ability to solve problems in real scenarios. This is consistent with human experience: larger language model parameters mean more text logic training samples and less model distillation. When faced with more complex logical reasoning tasks, these models can leverage more underlying knowledge and fundamental capabilities.

## 4.3 SCENARIO DIMENSION ANALYSIS

**The performance of LMMs in daily scenarios still has great room for improvement.** To observe the specific performance of different models in various scenarios, as shown in Figure 3, we calculated the accuracy of different models across different fields. We found that these 14 LMMs achieved good performance in Level 1 for the education scenario. The performance is more balanced in the architecture, housework, transport and social service scenarios. However, there are some shortcomings in the performance of sports scenarios, which we believe are closely related to the current training data of LMMs. At present, LMMs research groups focus more on achieving better training and testing levels using existing Internet text data and high-quality textbook data, but they neglect the improvement of datasets and capabilities in everyday life fields. This is where the MDI-Benchmark comes into play. We believe that the types of problems related to logical reasoning and the required background knowledge in the fields of sport and transport are richer and broader than those in architecture, resulting in increased problem difficulty and a significant gap in reasoning performance.

## 4.4 COMPLEXITY DIMENSION ANALYSIS

**Decreased performance with increased complexity.** As the complexity of the problems increases, the model's performance in every scenario noticeably decreases. The accuracy of answering questions in the same scenario can also change significantly for the same model. For instance, in the case of GPT-4o, the accuracy in the best-performing educational scenario dropped from 94.12 to 70.59. This highlights the significant impact of problem complexity on model performance.

**The complexity of questions presents a rich diversity in generalization when it comes to different scenarios.** To analyze the detailed performance of these LMMs across multiple levels, we create radar charts (Figure 4) that display the performance of 14 LMMs in various scenarios under Level 1 and Level 2. To illustrate macro performance changes due to varying problem complexity, we also generate statistics of performance variance and summation, plotting average and variance

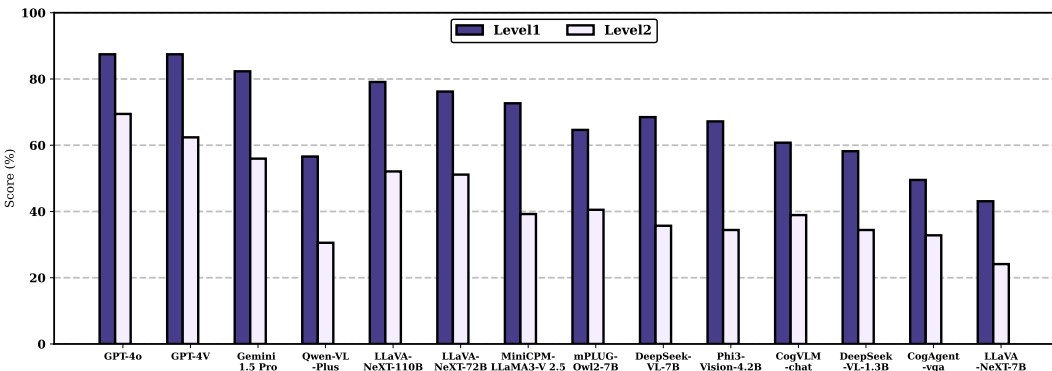

Figure 3: The average performance of different LMMs on different difficulty levels of the MDI-Benchmark.

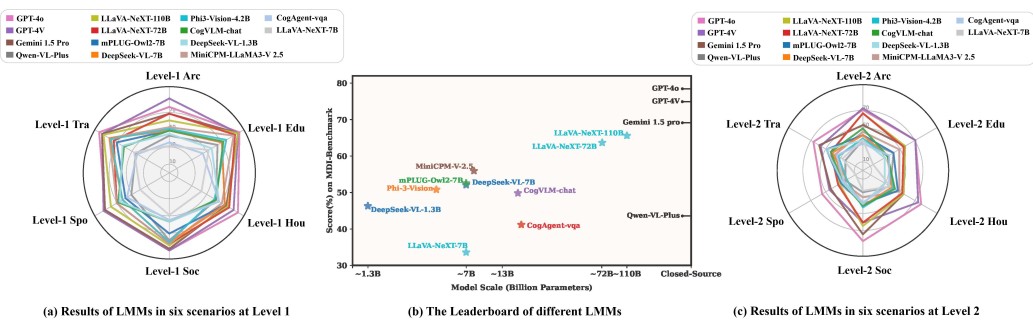

(a) Results of LMMs in six scenarios at Level 1     (b) The Leaderboard of different LMMs     (c) Results of LMMs in six scenarios at Level 2

Figure 4: Performance of the model at different difficulty levels and the overall performance results of the model under the score metric.

data on different axes to highlight macro trends (Figure 5). Generally, models with high averages and low variances exhibit better and more comprehensive capabilities.

We find that under Level 1, most models maintain relatively balanced performance—radar maps show a normal hexagon shape—with exceptions like CogAgent-vqa and LLaVA-NeXT-7B. Under Level 2, GPT-4o's variance increases significantly, with only the GPT series and Gemini 1.5 Pro maintaining balanced performance. Observing the radar maps, only the GPT series shows slight performance degradation, while other LMMs exhibit a steep decline in the sports scenario.

Compared to advanced closed-source LMMs, open-source LMMs require further research on specific daily life capabilities and complex problem scenarios to bridge the significant gap. Notably, LLaVA-NeXT-72B performs similarly to the optimal model LLaVA-NeXT-110B at Level 2 but with decreased variance, suggesting that effective distillation to achieve better performance with smaller parameters is a worthy area for further investigation.

We believe that the research community's lack of focus on enhancing LMMs datasets and capabilities in these areas, along with the diverse and extensive types of problems associated with logical reasoning and required background knowledge, is more pronounced compared to simpler tasks. This diversity results in significant gaps in the model's inference perfor-

| Model | Avg | old | middle-aged | young |
|---|---|---|---|---|
| *Closed-source* | | | | |
| **GPT-4o** | 79.74 | 77.94 | 78.43 | 82.84 |
| **GPT-4V** | 76.14 | 75.49 | 75.49 | 77.45 |
| **Gemini 1.5 Pro** | 70.26 | 70.10 | 68.63 | 72.06 |
| **Qwen-VL-Plus** | 44.28 | 41.67 | 40.20 | 50.98 |
| *Open-source* | | | | |
| **LLaVA-NeXT-110B** | 66.67 | 69.12 | 63.24 | 67.65 |
| **LLaVA-NeXT-72B** | 64.71 | 66.67 | 63.73 | 63.73 |
| **MiniCPM-LLaMA3-V 2.5** | 56.86 | 55.88 | 54.90 | 59.80 |
| **mPLUG-Owl2-7B** | 53.43 | 55.39 | 50.98 | 53.92 |
| **DeepSeek-VL-7B** | 52.94 | 53.43 | 51.96 | 53.43 |
| **Phi3-Vision-4.2B** | 51.63 | 53.43 | 49.02 | 52.45 |
| **CogVLM-chat** | 50.65 | 52.94 | 51.96 | 47.06 |
| **DeepSeek-VL-1.3B** | 47.06 | 49.02 | 39.71 | 52.45 |
| **CogAgent-vqa** | 41.83 | 44.12 | 42.65 | 38.73 |
| **LLaVA-NeXT-7B** | 34.15 | 37.75 | 33.82 | 30.88 |

Table 4: Performance of Various Models Across Different Age Groups. The best performance in each block is highlighted in blue and green.

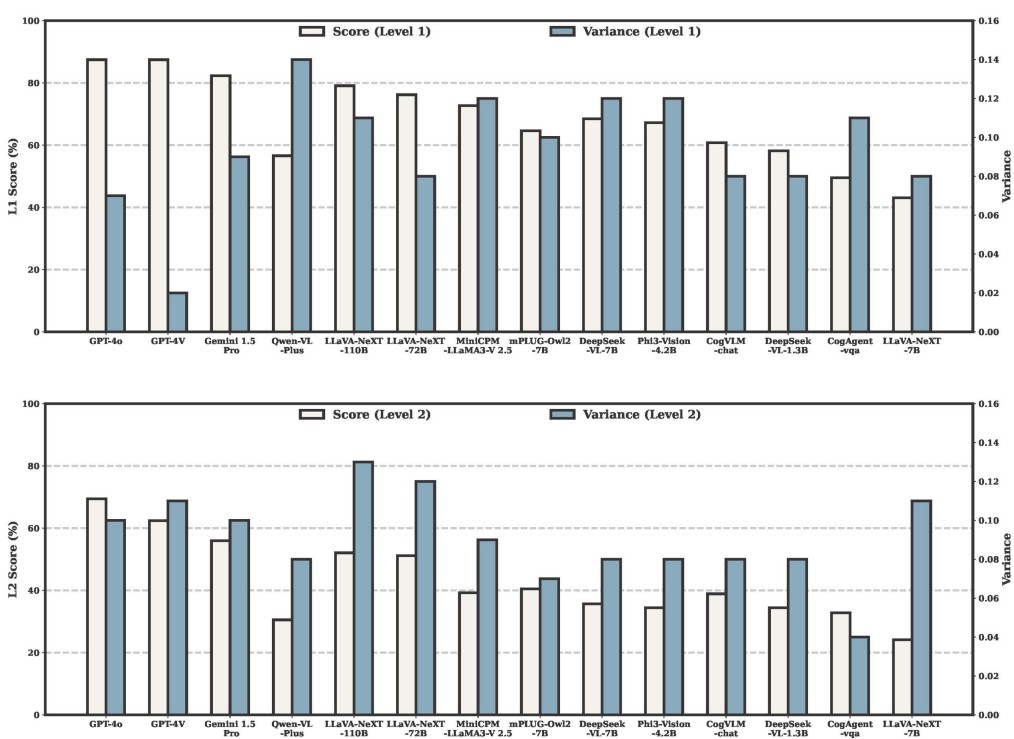

Figure 5: The average accuracy and variance of LLMs across six domains at Level 1 and Level 2

mance as the complexity of the problems increases. Therefore, further research is needed to address these gaps and improve LMM performance in complex problem scenarios.

## 4.5 AGE DIMENSION ANALYSIS

For a more direct and macro-level performance analysis, we only presented the average performance statistics in the main table, as shown in Table 4 , which primarily represents the performance of LMMs across three age stratification. Furthermore, we analyzed the model's performance in detail based on age groups and scenario dimensions, as shown in the Appendix C. We have the following observations.

**All the models to follow under the level evaluation dimensions, but there are differences in performance between different age.** As shown in Table 4, GPT-4o remains the top-performing model in the age dimension, demonstrating a performance advantage of 13 points over the highest-ranked open-source model and 35 points over the lowest-ranked closed-source model. This dominant performance in the age-stratified evaluation highlights GPT-4o's strong generalization ability and its leadership in daily use scenarios. However, when evaluating the model's capabilities from the perspective of the age dimension, it provides insights into the model's effectiveness across different groups in various real-world scenarios. Given the multitude of situations individuals encounter in daily life, a model's capabilities must be comprehensive to address diverse human needs. The observed decline in accuracy across age groups indicates that there is significant room for improvement in the overall performance of all models within this dimension. This finding underscores the need for further research focusing on age-related issues and highlights both the necessity and innovation of our work.

**Models exhibit insufficient overall generalization across different age dimensions.** As shown in Figure 6, we further visualize the model's performance across different age group, including old, middle-aged, young. By summing the model's results across age dimensions, we find that the old group achieves a total of 856.38, the middle-aged group 764.72, and the young group 902.94. This

distribution highlights the actual difficulty order of questions across age levels: middle-aged >old >young. In real-world scenarios, questions posed by middle-aged individuals tend to encompass more aspects and require greater logical reasoning and background knowledge than those from older or younger individuals. Therefore, multi-modal LMMs need to have robust and comprehensive capabilities to effectively handle such questions. GPT-4o demonstrates strong performance in this aspect, exhibiting smaller performance gaps across all three age-related categories. Interestingly, the Cog-series model, despite having the largest visual encoder, shows a noticeable performance drop in the young group, suggesting that its large visual encoder does not generalize as effectively as CLIP-ViT/L14.

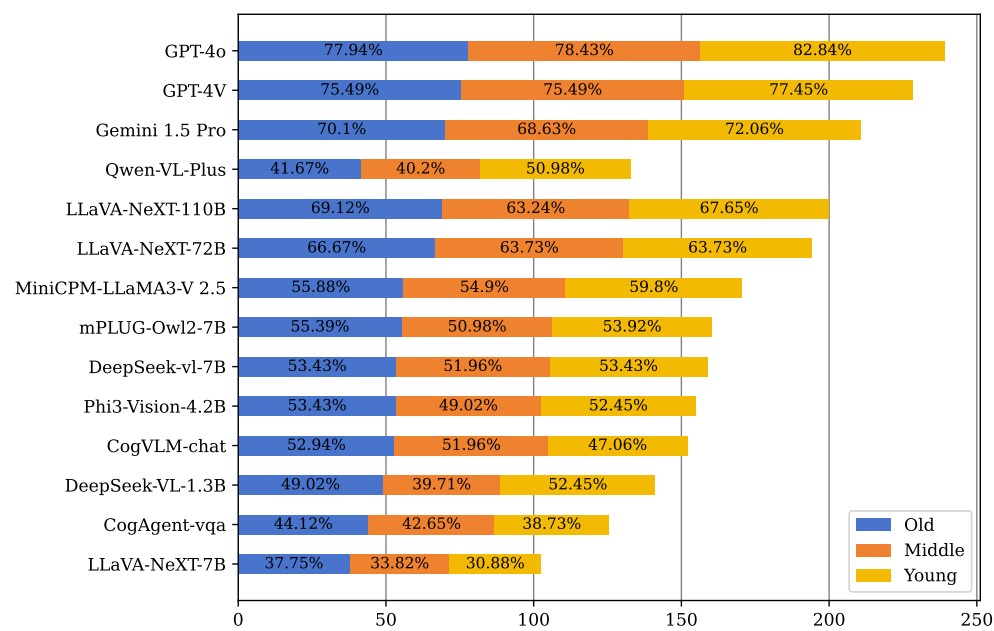

Figure 6: Performance of different LMMs across the age dimension.

In the age dimension, the scaling performance of language models is evident, but model compression shows great potential. We find that at each model layer, the model with the largest language model parameters achieved the best performance. Empirically, we believe that language models play a more important role in LMMs than visual encoders. Additionally, we are surprised to find that Phi3-Vision-4.2B exceed the macro performance of the closed-source model Qwen-VL-Plus using only about 4.2B parameters. This indicates that LMMs still have significant room for exploration in terms of model parameter compression.

## 5 CONCLUSION

In this paper, we propose the MDI-Benchmark, a tool designed to evaluate the capability of Large Multimodal Models (LMMs) in addressing real-world human demands within multi-dimensional scenarios. The MDI-Benchmark comprises over 500 images and 1.2k corresponding requirements, encompassing six major aspects of human life. Additionally, we introduce the concept of age stratification and sampling questions based on the needs of elderly, middle-aged, and young individuals to ensure comprehensive evaluation. Using the MDI-Benchmark, we evaluated 14 existing LMMs, revealing their performance preferences in different scenarios. While GPT-4o performed best across a variety of metrics, there were gaps in performance across all age groups and scenarios. Therefore, we suggest that future studies should focus on improving the adaptability of LMM to human needs and its ability to generalize across different domains and age groups. This will pave the way for the next generation of LMMs that can effectively meet human needs.

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

## LIMITATIONS

In this paper, we introduce MDI-Benchmark, the first benchmark to incorporate personalized preference requirements, represented by age, into LMM evaluation. However, there are still some limitations to this paper.

(1) Scope Coverage: Given the infinite possibilities of real-world scenarios, MDI-Benchmark cannot cover all domains. MDI-Benchmark focuses on 18 fine-grained subfields within 6 key domains.

(2) Task Format: To achieve automated evaluation, similar to many other benchmarks, we use multiple-choice questions as the sole task format in MDI-Benchmark.

(3) Data Scale: MDI-Benchmark consists of 500 meticulously hand-collected and processed images and 1,298 questions. How to automatically construct large-scale, high-quality customized preference data remains to be explored.

(4) Preference Dimensions: MDI-Benchmark selects the most common preference dimension, age, to evaluate LMM. We leave the exploration of other customized preference dimensions in future work.

## A    MORE DETAILS ON EXPERIMENT SETUP

### A.1    DETAILS OF THE EVALUATED MODELS

Table 5 shows the release times and model sources of the LMMs we evaluated at MDI-Benchmark.

Table 5: The release time and model source of LMMs used in MDI-Benchmark

| Model | Release Time | Source |
|---|---|---|
| GPT-4o (OpenAI, 2024) | 2024-05 | https://gpt4o.ai/ |
| GPT-4V (OpenAI, 2023) | 2024-04 | https://openai.com/index/gpt-4v-system-card/ |
| Gemini 1.5 Pro (Team et al., 2023) | 2024-05 | https://deepmind.google/technologies/gemini/pro/ |
| Qwen-VL-Plus (Bai et al., 2023) | 2024-01 | https://huggingface.co/spaces/Qwen/Qwen-VL-Plus/ |
| LLaVA-NeXT-110B (Liu et al., 2024a) | 2024-05 | https://huggingface.co/lmms-lab/llava-next-110b/ |
| LLaVA-NeXT-72B (Liu et al., 2024a) | 2024-05 | https://huggingface.co/lmms-lab/llava-next-72b/ |
| MiniCPM-LLaMA3-V 2.5 (Hu et al., 2024) | 2024-05 | https://huggingface.co/openbmb/MiniCPM-Llama3-V-2_5/ |
| mPLUG-Owl2-7B (Ye et al., 2023) | 2023-11 | https://huggingface.co/MAGAer13/mplug-owl2-llama2-7b |
| DeepSeek-VL-7B (Lu et al., 2024a) | 2024-03 | https://huggingface.co/deepseek-ai/deepseek-vl-7b-chat/ |
| Phi3-Vision-4.2B (Abdin et al., 2024) | 2024-05 | https://huggingface.co/microsoft/Phi-3-vision-128k-instruct/ |
| CogVLM-chat (Wang et al., 2023) | 2023-12 | https://huggingface.co/THUDM/cogvlm-chat-hf/ |
| DeepSeek-VL-1.3B (Lu et al., 2024a) | 2024-03 | https://huggingface.co/deepseek-ai/deepseek-vl-1.3b-chat/ |
| CogAgent-vqa (Hong et al., 2023) | 2023-12 | https://huggingface.co/THUDM/cogagent-vqa-hf/ |
| LLaVA-NeXT-7B (Liu et al., 2024a) | 2024-03 | https://huggingface.co/llava-hf/llava-v1.6-vicuna-7b-hf/ |

## B    MORE DETAIL ON MDI-BENCHMARK

### B.1    EXAMPLE OF SCENARIO DIMENSION

In this section, we present a selection of images from the MDI-Benchmark for visual demonstration purposes.

1. **Architecture**: Including house planning, work scenes, measuring, etc. As shown in Figure 7.

2. **Education**: Including campus facilities, studying activities, teaching, etc. As shown in Figure 8.

3. **Housework**: Including home arrangements, housework activities, household appliances, etc. As shown in Figure 9.

4. **Social service**: Including travel, shopping, communal facilities, etc. As shown in Figure 10.

5. **Sport:** Including ball sports, racing sports, powerlifting, etc. As shown in Figure 11.

6. **Transport:** including signpost, rail transit, airport, etc. As shown in Figure  12.

**Architecture:**

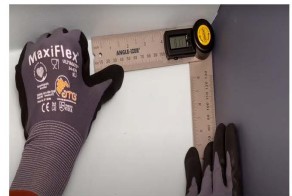

Q: What color is used to distinguish bath in the room plan?
A. red
B. yellow
C. black
D. blue
GT: D

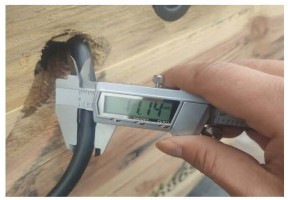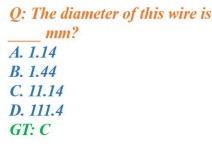

Q: The diameter of this wire is _____ mm?
A. 1.14
B. 1.44
C. 11.14
D. 111.4
GT: C

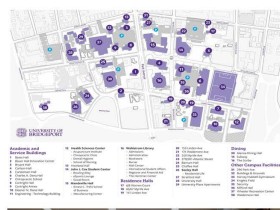

Q: The Angle of this corner is _____ °
A. 54.1
B. 79.5
C. 64.1
D. 81.7
GT: D

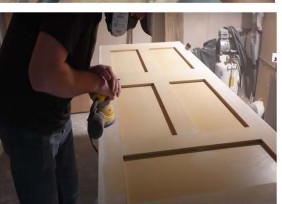

Q: What object is the worker polishing?
A. door
B. window
C. desk
D. floor
GT: A

Figure 7: Examples of Architecture Scenario.

**Education:**

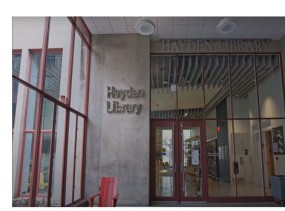

Q: According to the map of the university, how many parking lots are there?
A. 10
B. 9
C. 8
D. 7
GT: D

Q: What's the writing on the wall?
A. Hayden Library
B. Heyden Library
C. Haryden Library
D. Haydan Library
GT: A

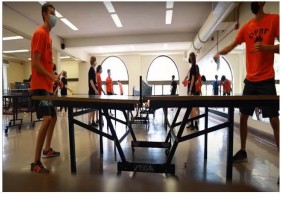

Q: What sports are they playing?
A. billiards
B. air hockey
C. foosball
D. table tennis
GT: D

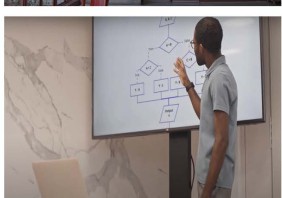

Q: What is being displayed on the whiteboard?
A. algorithm flow
B. personnel placement
C. exam results
D. school calendar
GT: A

Figure 8: Examples of Education Scenario.

**Housework:**

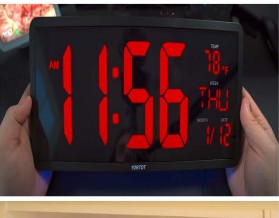

Q: What day is it today?
A. Friday
B. Sunday
C. Thursday
D. Monday
GT: C

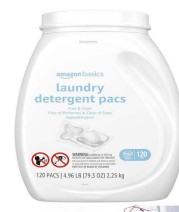

Q: This one contains _____ individual packaging?
A. 60
B. 120
C. 100
D. 20
GT: B

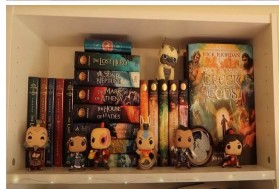

Q: How many dolls are there on the shelf?
A. 5
B. 7
C. 9
D. 15
GT: B

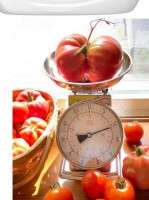

Q: What is being measured?
A. potato
B. onion
C. tomato
D. strawberry
GT: C

Figure 9: Examples of Housework Scenario.

*Social service:*

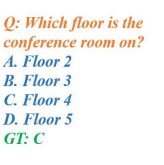

*Q: What's the signature of this shop?*
*A. burger*
*B. barbecue*
*C. taco*
*D. waffle*
*GT: D*

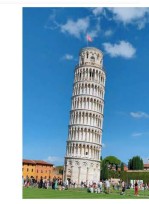

*Q: Nvidia shares closed yesterday at $____?*
*A. 914.12*
*B. 884.98*
*C. 971.13*
*D. 925.61*
*GT: D*

*Q: Which floor is the conference room on?*
*A. Floor 2*
*B. Floor 3*
*C. Floor 4*
*D. Floor 5*
*GT: C*

*Q: Which famous building is shown in the picture?*
*A. The Sistine Chapel*
*B. Pompeii*
*C. Torre pendente di Pisa*
*D. Palazzo Ducalculatee*
*GT: C*

Figure 10: Examples of Social Service.

*Sport:*

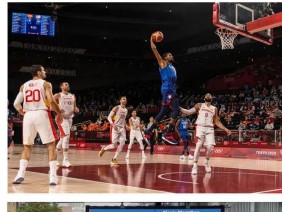

*Q: Who's this guy about to dunk?*
*A. Kevin Durant*
*B. Russell Westbrook*
*C. James Harden*
*D. Lebron James*
*GT: A*

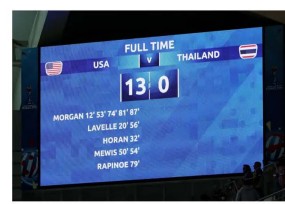

*Q: How many goals did the United States score?*
*A. 0*
*B. 13*
*C. 10*
*D. 3*
*GT: B*

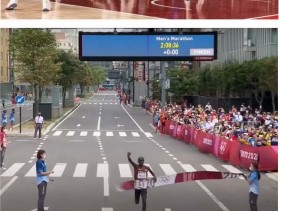

*Q: What was the athlete's final time in the race?*
*A. 2:05:21*
*B. 2:01:16*
*C. 2:03:88*
*D. 2:08:36*
*GT: D*

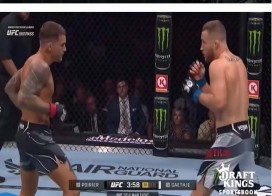

*Q: Which of these two players is ranked higher?*
*A. GAETHJE*
*B. POIRIER*
*C. uncertain*
*GT: B*

Figure 11: Examples of Sport Scenario.

*Transport:*

*Q: How many flights are the Flight closing now?*
*A. 1*
*B. 2*
*C. 3*
*D. 4*
*GT: C*

*Q: Which bus stops at this station during the day?*
*A. 63*
*B. 363*
*C. 36*
*D. 663*
*GT: A*

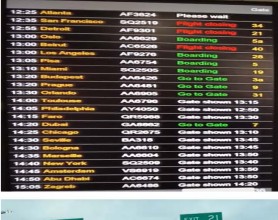

*Q: What is the exit number ahead of the road?*
*A. 19*
*B. 23*
*C. 21*
*D. 8*
*GT: C*

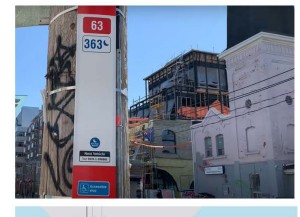

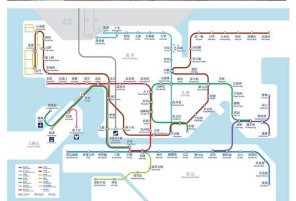

*Q: How many stations does the Tsuen Wan Line pass through?*
*A. 15*
*B. 16*
*C. 17*
*D. 18*
*GT: B*

Figure 12: Examples of Transport Scenario.

## B.2 EXAMPLE OF PROBLEM COMPLEXITY DIMENSION

In this section, we present questions of varying difficulties across six scenario dimensions, as shown in Figures 13 to Figure 18. It is evident that Level 1 questions are relatively simple, while Level 2 questions require LMMs to use more advanced abilities to answer.

*Architecture:*

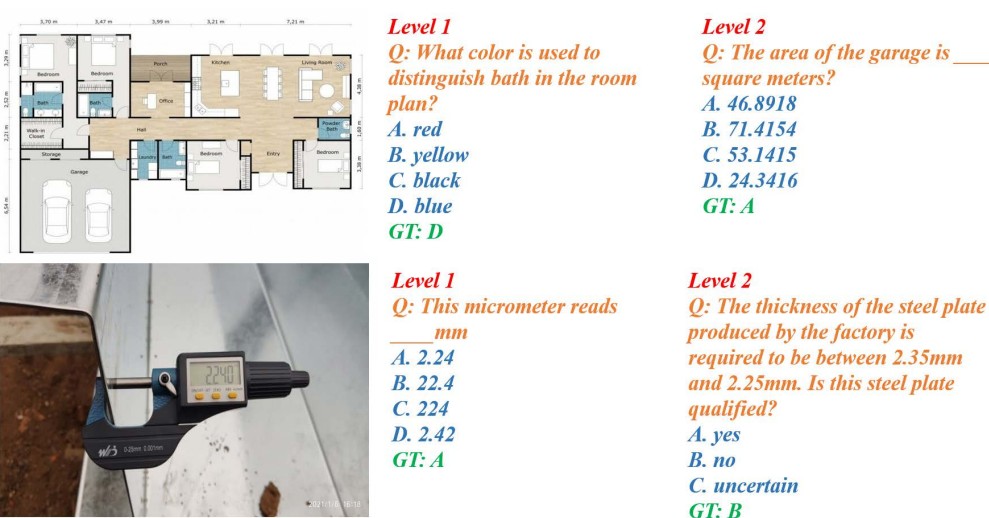

Figure 13: Examples of Architecture Scenario Questions.

*Education:*

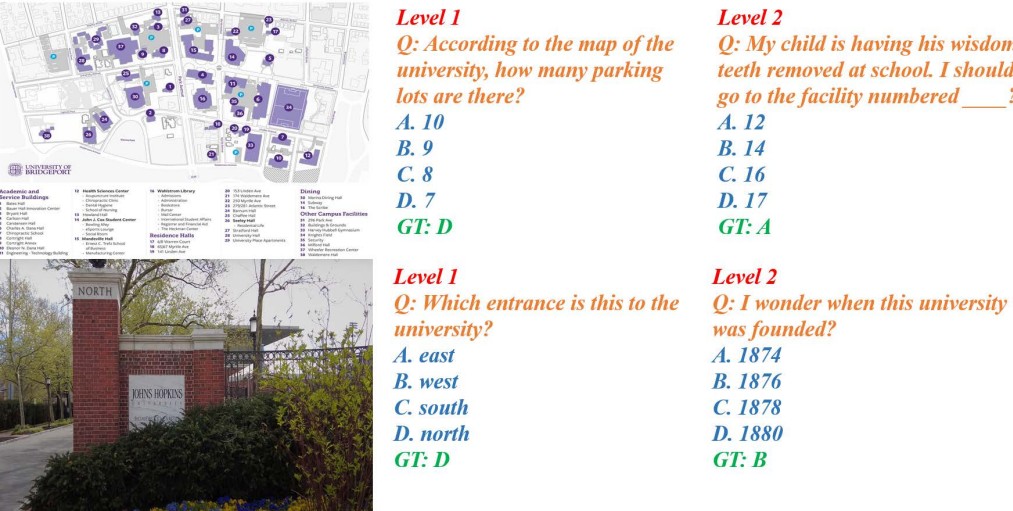

Figure 14: Examples of Education Scenario Questions.

*Housework:*

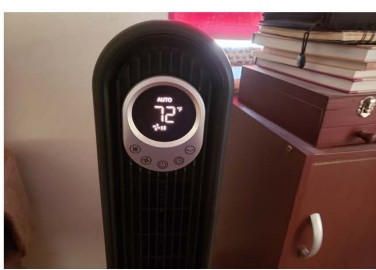

**Level 1**
Q: What's the temperature set on this fan?
A. 62 ℉
B. 52 ℉
C. 72 ℉
D. 82 ℉
GT: C

**Level 2**
Q: I think the current wind is a little low, which button should I use to adjust the wind(Select from left to right)?
A. 1
B. 2
C. 3
D. 4
E. 5
GT: B

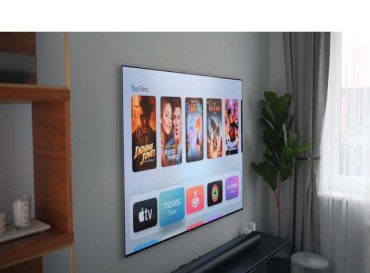

**Level 1**
Q: How many movies are available on TV right now?
A. 2
B. 3
C. 4
D. 5
GT: D

**Level 2**
Q: I'm a big fan of DC Comics, which movie would you recommend I watch on the list of movies shown on TV? (Select from left to right)?
A. 1
B. 2
C. 3
D. 4
E. 5
GT: D

Figure 15: Examples of Housework Scenario Questions.

*Social service:*

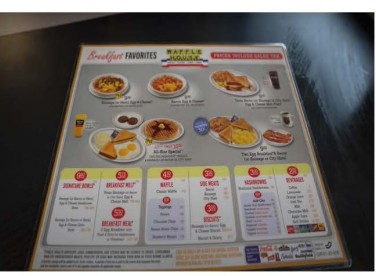

**Level 1**
Q: What's the signature of this shop?
A. burger
B. barbecue
C. taco
D. waffle
GT: D

**Level 2**
Q: The doctor asked me to limit my breakfast to around 600 calories. What should I order after I order a classic waffle and bacon?
A. Lemonade
B. Apple Juice
C. Chocolate Milk
D. Coffee
GT: D

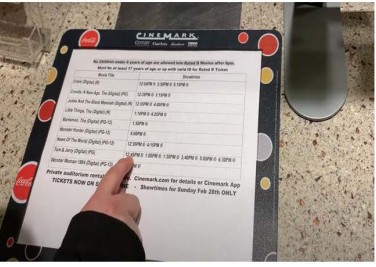

**Level 1**
Q: How many movies are on today?
A. 7
B. 8
C. 9
D. 10
GT: C

**Level 2**
Q: We are a family of four with two 5-year-olds and it is 6:15 PM, which movie am I fit to buy?
A. Monster Hunter
B. Wonder Woman 1984
C. Tom and Jerry
D. Little Things
GT: D

Figure 16: Examples of Social Service Scenario Questions.

**Sport:**

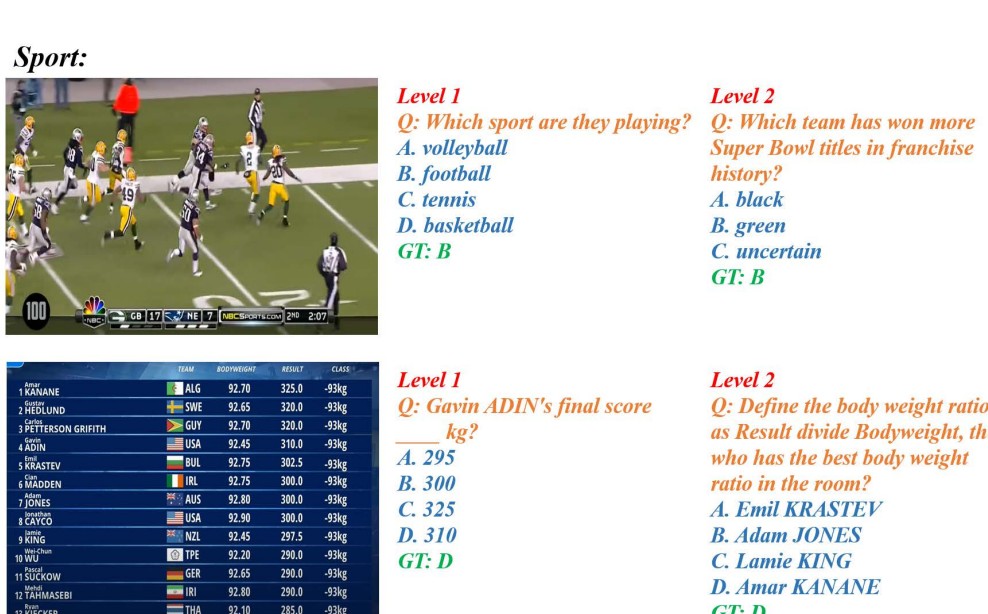

*Level 1*
*Q: Which sport are they playing?*
*A. volleyball*
*B. football*
*C. tennis*
*D. basketball*
*GT: B*

*Level 2*
*Q: Which team has won more Super Bowl titles in franchise history?*
*A. black*
*B. green*
*C. uncertain*
*GT: B*

*Level 1*
*Q: Gavin ADIN's final score _____ kg?*
*A. 295*
*B. 300*
*C. 325*
*D. 310*
*GT: D*

*Level 2*
*Q: Define the body weight ratio as Result divide Bodyweight, then who has the best body weight ratio in the room?*
*A. Emil KRASTEV*
*B. Adam JONES*
*C. Lamie KING*
*D. Amar KANANE*
*GT: D*

Figure 17: Examples of Sport Scenario Questions.

**Transport:**

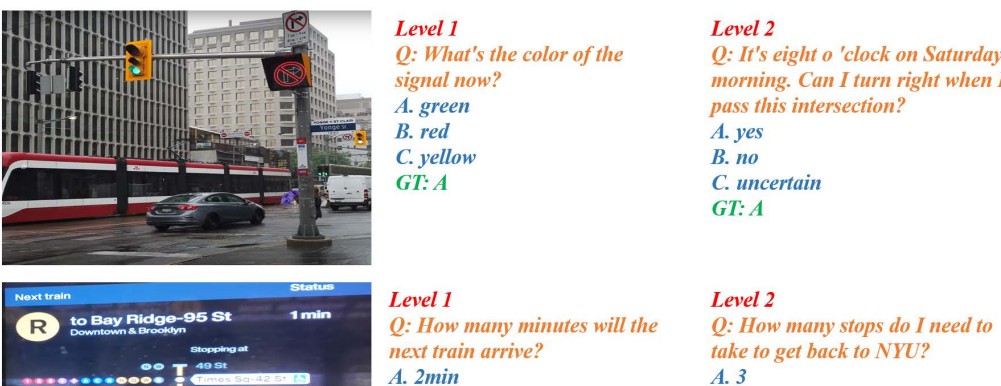

*Level 1*
*Q: What's the color of the signal now?*
*A. green*
*B. red*
*C. yellow*
*GT: A*

*Level 2*
*Q: It's eight o 'clock on Saturday morning. Can I turn right when I pass this intersection?*
*A. yes*
*B. no*
*C. uncertain*
*GT: A*

*Level 1*
*Q: How many minutes will the next train arrive?*
*A. 2min*
*B. 3min*
*C. 5min*
*D. 1min*
*GT: D*

*Level 2*
*Q: How many stops do I need to take to get back to NYU?*
*A. 3*
*B. 4*
*C. 5*
*D. 6*
*GT: C*

Figure 18: Examples of Transport Scenario Question.

## B.3 Example of Age Dimention

In this section, we have sampled various concerns and issues from people across three different age groups within the six major scenarios. These concerns have been categorized by scenario and are visually presented in Figures 19 through 24.

*Architecture:*

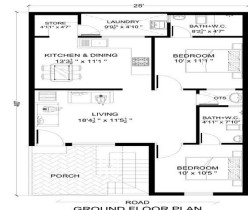

**Old_Q**
*We would like to plant some flowers and plants in the porch. How large surface foot is the area of this porch?*
*A. 481*
*B. 335*
*C. 435*
*D. 378*
*GT: C*

**Mid_Q**
*According to the equipment recommended in the house plan, is the bathroom in our home more suitable for installing a shower or a bathtub?*
*A. shower*
*B. bathtub*
*C. uncertain*
*GT: A*

**Young_Q**
*I want a desk in my smaller bedroom. There's already a closet on the south side. Where in the room should I put it?*
*A. East*
*B. South*
*C. West*
*D. North*
*GT: C*

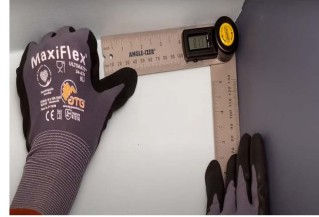

**Old_Q**
*Can the gloves I purchased be used to replace wires?*
*A. yes*
*B. no*
*C. uncertain*
*GT: C*

**Mid_Q**
*I want to buy a bookshelf with a square bottom and put it in the corner. Can the bookshelf fit neatly into the corner?*
*A. yes*
*B. no*
*C. uncertain*
*GT: B*

**Young_Q**
*Besides measuring angles, what else can this tool measure?*
*A. length*
*B. temperature*
*C. pressure*
*D. uncertain*
*GT: A*

Figure 19: Example of Architecture Scenario Age Questions.

*Education:*

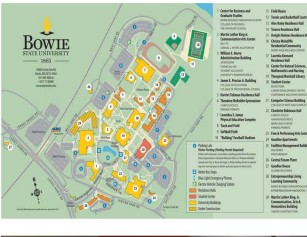

**Old_Q**
*Is the campus map showing the location of accessible seating?*
*A. yes*
*B. no*
*C. uncertain*
*GT: B*

**Mid_Q**
*My child's dorm is located in Building 17, my Tesla needs to be charged, which parking lot is best for me to wait for my child?*
*A. Q parking lot*
*B. P parking lot*
*C. H parking lot*
*D. G parking lot*
*GT: B*

**Young_Q**
*I am now going to a career planning course, which facility should I go to?*
*A. Building 20*
*B. Building 2*
*C. Building 22*
*D. Building 27*
*GT: A*

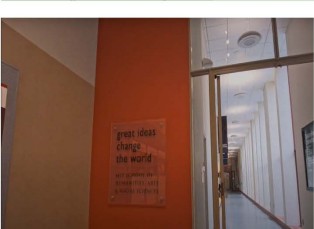

**Old_Q**
*How many years was this school founded?*
*A. 1807*
*B. 1861*
*C. 1844*
*D. 1897*
*GT: B*

**Mid_Q**
*I would like to browse some information about this college. What is the official website of this college?*
*A. https://www.ll.mit.edu/*
*B. https://science.mit.edu/*
*C. https://mitsloan.mit.edu/*
*D. https://shass.mit.edu/*
*GT: D*

**Young_Q**
*I came to visit the school. Which of the following subjects is not part of the curriculum offered by the School?*
*A. art*
*B. social science*
*C. social work*
*D. agronomy*
*GT: D*

Figure 20: Example of Education Scenario Age Questions.

**Housework:**

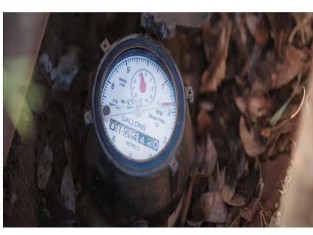

**Old_Q**
How many gallons does the meter read in black?
A. 280
B. 320
C. 360
D. 420
GT: D

**Mid_Q**
The last time I checked the watch, the read 0163390. We know that the water charge per gallon is $0.003. How much do I need to pay this time?
A. $1.23
B. $5.98
C. $3.09
D. $4.07
GT: C

**Young_Q**
What does this watch do?
A. Measure household electricity usage
B. Measure household water use
C. Measure household gas usage
D. uncertain
GT: B

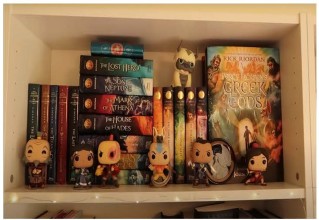

**Old_Q**
Who is the author of the first book on the right?
A. Rick Riordan
B. Greek
C. Jackson
D. J. K. Rowling
GT: A

**Mid_Q**
My child's birthday is coming, I want to send him another small decoration like this, which brand should I go to buy?
A. Bandai
B. Funko pop
C. Mattel
D. LEGO
GT: B

**Young_Q**
Of these toys on the bookcase, which one is the last airbender(Count from left to right)?
A. 1
B. 2
C. 4
D. 6
GT: C

Figure 21: Example of Housework Scenario Age Questions.

**Social service:**

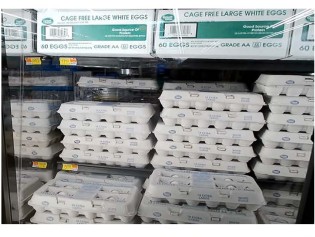

**Old_Q**
My doctor recommends that I eat two eggs a day, so do I buy one small package to cover my intake for two weeks?
A. yes
B. no
C. uncertain
GT: B

**Mid_Q**
How much cheaper is the larger package than the smaller package for the same 60 eggs?
A. $1.64
B. $9.24
C. $5.66
D. $2.31
GT: D

**Young_Q**
By what agency is the grade of this product confirmed?
A. WFP
B. FAO
C. IFAD
D. USDA
GT: D

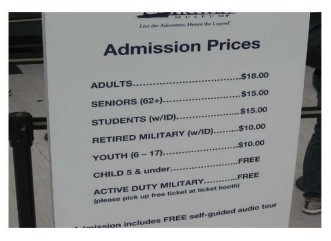

**Old_Q**
I am 70 years old this year, can I enjoy free admission?
A. yes
B. no
C. uncertain
GT: B

**Mid_Q**
Now there are four adults in our tour group, one of whom is a retired soldier, three students and two infants. How much does the ticket cost?
A. $109
B. $99
C. $129
D. $159
GT: A

**Young_Q**
What exhibits can I see in this museum? 1.Ship model 2.Ship construction technology 3.Maritime history 4.Religious belief
A. 1 and 2 and 3
B. 2 and 3 and 4
C. 1 and 2 and 4
D. 1 and 3 and 4
GT: A

Figure 22: Example of Social Service Scenario Age Questions.

*Sport:*

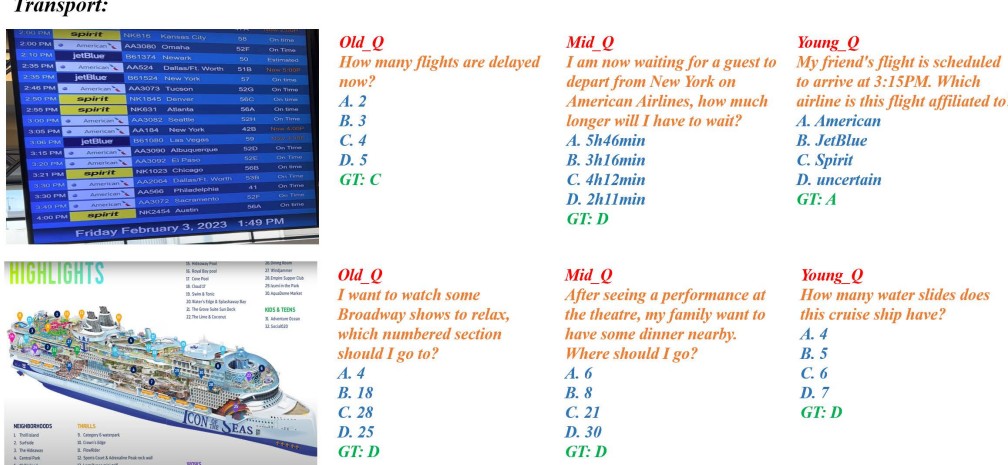

**Old_Q**
*In what year was the sport added to the Olympic program?*
*A. 1912*
*B. 1956*
*C. 1988*
*D.1924*
*GT: A*

**Mid_Q**
*What caliber bullets were used by these athletes?*
*A. .22 BB*
*B. .22 Magnum*
*C. .22 LR*
*D. uncertain*
*GT: C*

**Young_Q**
*What was the final score for the first place finisher?*
*A. 107.6*
*B. 104.6*
*C. 104.5*
*D. 105.6*
*GT: D*

**Old_Q**
*Who is the referee of this match?*
*A. Herb Dean*
*B. Marc Goddard*
*C. Jason Herzog*
*D. Mark Smith*
*GT: A*

**Mid_Q**
*Under what weight did they fight this fight?*
*A. Featherweight*
*B. Lightweight*
*C. Welterweight*
*D. Bantamweight*
*GT: B*

**Young_Q**
*How long will the whole match last?*
*A. 5min*
*B. 15min*
*C. 25min*
*D. 35min*
*GT: C*

Figure 23: Example of Sport Scenario Age Questions.

*Transport:*

**Old_Q**
*How many flights are delayed now?*
*A. 2*
*B. 3*
*C. 4*
*D. 5*
*GT: C*

**Mid_Q**
*I am now waiting for a guest to depart from New York on American Airlines, how much longer will I have to wait?*
*A. 5h46min*
*B. 3h16min*
*C. 4h12min*
*D. 2h11min*
*GT: D*

**Young_Q**
*My friend's flight is scheduled to arrive at 3:15PM. Which airline is this flight affiliated to?*
*A. American*
*B. JetBlue*
*C. Spirit*
*D. uncertain*
*GT: A*

**Old_Q**
*I want to watch some Broadway shows to relax, which numbered section should I go to?*
*A. 4*
*B. 18*
*C. 28*
*D. 25*
*GT: D*

**Mid_Q**
*After seeing a performance at the theatre, my family want to have some dinner nearby. Where should I go?*
*A. 6*
*B. 8*
*C. 21*
*D. 30*
*GT: D*

**Young_Q**
*How many water slides does this cruise ship have?*
*A. 4*
*B. 5*
*C. 6*
*D. 7*
*GT: D*

Figure 24: Example of Transport Scenario Age Questions.

## C  MORE DETAILS ON EXPERIMENT RESULTS

We present the performance of models across different age groups in Table 6.

Table 6: Performance of models across different age groups. The best performance in each block is highlighted in blue and green.

| Model | Avg Old | Mid | Young | Arc Old | Mid | Young | Edu Old | Mid | Young | Hou Old | Mid | Young | Soc Old | Mid | Young | Spo Old | Mid | Young | Tra Old | Mid | Young |
|---|---|---|---|---|---|---|---|---|---|---|---|---|---|---|---|---|---|---|---|---|---|
| *Closed-source* | | | | | | | | | | | | | | | | | | | | | |
| GPT-4o | 77.94 | 78.43 | 82.84 | 79.41 | 67.65 | 73.53 | 85.29 | 79.41 | 82.35 | 82.35 | 82.35 | 91.18 | 88.24 | 79.41 | 91.18 | 64.71 | 76.47 | 70.59 | 67.65 | 85.29 | 88.24 |
| GPT-4V | 75.49 | 75.49 | 77.45 | 79.41 | 76.47 | 82.35 | 82.35 | 76.47 | 85.29 | 76.47 | 85.29 | 79.41 | 76.47 | 73.53 | 76.47 | 67.65 | 61.76 | 70.59 | 70.59 | 79.41 | 70.59 |
| Gemini 1.5 Pro | 70.10 | 68.63 | 72.06 | 58.82 | 47.06 | 76.47 | 73.53 | 79.41 | 70.59 | 67.65 | 64.71 | 64.71 | 85.29 | 70.59 | 88.24 | 55.88 | 67.65 | 70.59 | 79.41 | 82.35 | 61.76 |
| Qwen-VL-Plus | 41.67 | 40.20 | 50.98 | 38.24 | 32.35 | 47.06 | 44.12 | 52.94 | 61.76 | 50.00 | 38.24 | 61.76 | 50.00 | 47.06 | 58.82 | 32.35 | 38.24 | 41.18 | 35.29 | 32.35 | 35.29 |
| *Open-source* | | | | | | | | | | | | | | | | | | | | | |
| LLaVA-NeXT-110B | 69.12 | 63.24 | 67.65 | 73.53 | 52.94 | 64.71 | 76.47 | 76.47 | 70.59 | 70.59 | 67.65 | 61.76 | 76.47 | 64.71 | 82.35 | 50.00 | 55.88 | 58.82 | 67.65 | 61.76 | 67.65 |
| LLaVA-NeXT-72B | 66.67 | 63.73 | 63.73 | 73.53 | 58.82 | 70.59 | 73.53 | 73.53 | 67.65 | 67.65 | 67.65 | 64.71 | 73.53 | 61.76 | 79.41 | 52.94 | 55.88 | 47.06 | 58.82 | 64.71 | 52.94 |
| MiniCPM-LLaMA3-V 2.5 | 55.88 | 54.90 | 59.80 | 50.00 | 44.12 | 52.94 | 64.71 | 67.65 | 70.59 | 58.82 | 52.94 | 67.65 | 55.88 | 50.00 | 64.71 | 47.06 | 50.00 | 50.00 | 58.82 | 64.71 | 52.94 |
| mPLUG-Owl2-7B | 55.39 | 50.98 | 53.92 | 47.06 | 38.24 | 50.00 | 73.53 | 44.12 | 50.00 | 58.82 | 64.71 | 58.82 | 58.82 | 52.94 | 52.94 | 38.24 | 47.06 | 47.06 | 55.88 | 58.82 | 64.71 |
| DeepSeek-VL-7B | 53.43 | 51.96 | 53.43 | 41.18 | 41.18 | 52.94 | 61.76 | 50.00 | 44.12 | 55.88 | 55.88 | 58.82 | 61.76 | 44.12 | 76.47 | 41.18 | 52.94 | 32.35 | 58.82 | 67.65 | 55.88 |
| Phi3-Vision-4.2B | 53.43 | 49.02 | 52.45 | 44.12 | 41.18 | 47.06 | 58.82 | 52.94 | 52.94 | 52.94 | 44.12 | 55.88 | 64.71 | 58.82 | 61.76 | 50.00 | 38.24 | 38.24 | 50.00 | 58.82 | 58.82 |
| CogVLM-chat | 52.94 | 51.96 | 47.06 | 44.12 | 58.82 | 44.12 | 61.76 | 50.00 | 47.06 | 52.94 | 55.88 | 50.00 | 50.00 | 50.00 | 47.06 | 41.18 | 52.94 | 50.00 | 67.65 | 44.12 | 44.12 |
| DeepSeek-VL-1.3B | 49.02 | 39.71 | 52.45 | 41.18 | 29.41 | 50.00 | 50.00 | 32.35 | 47.06 | 50.00 | 47.06 | 47.06 | 58.82 | 35.29 | 50.00 | 29.41 | 52.94 | 58.82 | 64.71 | 41.18 | 61.76 |
| CogAgent-vqa | 44.12 | 42.65 | 38.73 | 32.35 | 41.18 | 26.47 | 38.24 | 47.06 | 35.29 | 50.00 | 52.94 | 50.00 | 52.94 | 35.29 | 50.00 | 41.18 | 47.06 | 35.29 | 50.00 | 32.35 | 35.29 |
| LLaVA-NeXT-7B | 37.75 | 33.82 | 30.88 | 32.35 | 32.35 | 35.29 | 35.29 | 38.24 | 26.47 | 44.12 | 47.06 | 29.41 | 41.18 | 26.47 | 41.18 | 32.35 | 26.47 | 14.71 | 41.18 | 32.35 | 38.24 |

## D  CORRECT RESPONDS FROM GPT-4O

In view of GPT-4o's leading position in each scene and age dimension, we selected the correct answers and their reasoning processes for each scenario to display. The results are shown in Figures 25 through 30.

*Architecture:*

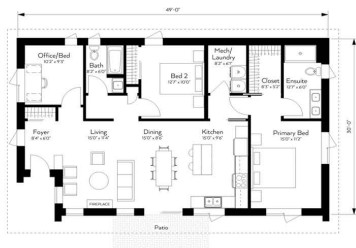

*Q: Now I want to buy a dining table, please judge which of the following dining table is the right size?*
*A. 15'1'' × 9'0''*
*B. 6'5'' × 4'5''*
*C. 11'3'' × 10'3''*
*D. 12'5'' × 11'7''*
*GT: B*

*GPT-4o:*
*<Reason>*
*Option A (15'1" x 9'0") is too large as it exceeds the room dimensions.*
*Option B (6'5 x 4'5") is a good size and will fit comfortably within the room.*
*Option C (11'3" x 10'3") is too wide for the room's width of 8'6".*
*Option D (12'5" x 11'7") is also too wide for the room's width of 8'6".*
*<Answer>B. 6'5'' × 4'5''*

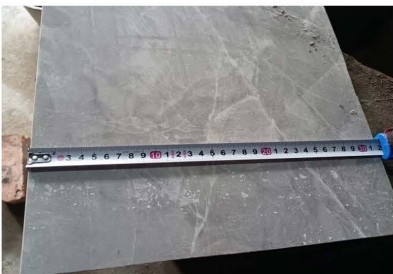

*Q: In the decoration process, I need to lay the floor with a length of 1.5 meters, and at least how many such floor tiles are needed?*
*A. 3*
*B. 4*
*C. 5*
*D. 6*
*GT: C*

*GPT-4o:*
*<Reason>The image shows a floor tile with a length of approximately 30 centimeters (0.3 meters). To cover a length of 1.5 meters, we need to determine how many such tiles are required. We can calculate this by dividing the total length needed (1.5 meters) by the length of one tile (0.3 meters).*
*1.5 meters / 0.3 meters per tile = 5 tiles*
*Therefore, at least 5 tiles are needed to cover the length of 1.5 meters.*
*<Answer>C. 5*

Figure 25: Example of GPT-4o Architecture Scenario Correct Answers.

*Education:*

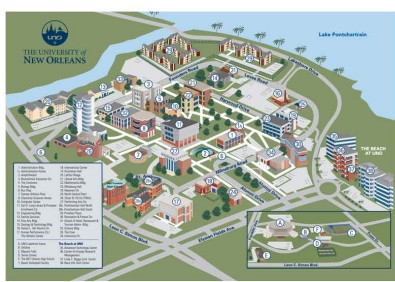

*Q: The school is giving us a physical fitness test now. We should go to the building numbered ____?*
*A. 17*
*B. 16*
*C. 22*
*D. 7*
*GT: A*

*GPT-4o:*
*<Reason>To determine which building to go to for the physical fitness test, I need to identify the building associated with physical fitness or related activities. On the map, building number 17 is labeled as "The Human Performance Center," which is likely the location for physical fitness activities.*
*<Answer>A. 17*

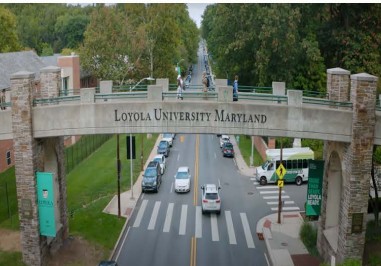

*Q: How many years has this school been established?*
*A. 132*
*B. 188*
*C. 172*
*D. 203*
*GT: C*

*GPT-4o:*
*<Reason>The image shows a bridge with the name "Loyola University Maryland" and a sign that reads "Loyola College 1852." To determine how many years the school has been established, I will subtract 1852 from the current year, 2023. 2023 - 1852 = 171 years. Since the closest option to 171 is 172, I will choose that.*
*<Answer>C. 172*

Figure 26: Example of GPT-4o Education Scenario Correct Answers.

*Housework:*

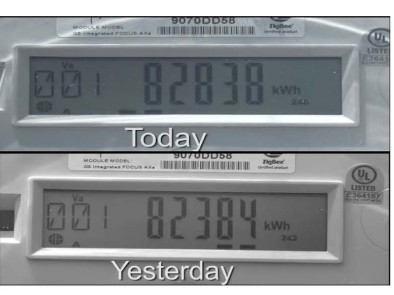

*Q: The electricity consumption from yesterday to today is ____kwh?*
*A. 454*
*B. 317*
*C. 762*
*D. 245*
*GT: A*

*GPT-4o:*
*<Reason>To determine the electricity consumption from yesterday to today. The reading for today is 82838 kWh, and the reading for yesterday is 82384 kWh. The difference between these two readings will give the electricity consumption.*
*Calculation:*
*82838 kWh (Today) - 82384 kWh (Yesterday) = 454 kWh*
*<Answer>A.454*

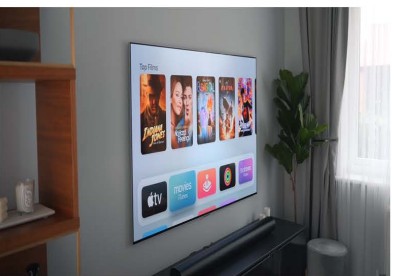

*Q: I'm a big fan of DC Comics, which movie would you recommend I watch on the list of movies shown on TV? (Select from left to right)?*
*A. 1*
*B. 2*
*C. 3*
*D. 4*
*E. 5*
*GT: D*

*GPT-4o:*
*<Reason>As a fan of DC Comics, you would likely enjoy a movie that features characters or stories from the DC universe. Among the movies shown on the TV, the fourth movie is "The Flash," which is a DC Comics character. Therefore, I would recommend you watch the fourth movie.*
*<Answer>D. 4*

Figure 27: Example of GPT-4o Housework Scenario Correct Answers.

*Social service:*

Q: My classmates and I came to this restaurant. Tina need 1 large tuna, Ben and Tom need 2 small spicy Italian and I want 1 large egg and cheese. How much will it cost?
A. $33.36
B. $43.36
C. $63.36
D. $23.36
GT: D

GPT-4o:
<Reason>
- Tina needs 1 large tuna, which costs $7.99.
- Ben and Tom need 2 small spicy Italian sandwiches, each costing $4.49. Therefore, 2 small spicy Italian sandwiches cost $4.49 * 2 = $8.98.
- I want 1 large egg and cheese, which costs $6.39.
Adding these together:
$7.99 (large tuna) + $8.98 (2 small spicy Italian) + $6.39 (large egg and cheese) = $23.36.
<Answer>D. $23.36

Q: What famous movie was shot here?
A. Gone with the Wind
B. Transformers
C. hangover
D. Titanic
GT: B

GPT-4o:
<Reason>The Griffith Observatory is a well-known landmark in Los Angeles, California. It has been featured in numerous films. Among the options provided, "Transformers" is the movie that includes scenes shot at the Griffith Observatory.
<Answer>B. Transformers

Figure 28: Example of GPT-4o Social Service Scenario Correct Answers.

*Sport:*

Q: Who scored the most goals in the game ?
A. RAPINOE
B. MEWIS
C. MORGAN
D. LAVELLE
GT: C

GPT-4o:
<Reason>I will analyze the scoreboard to determine which player scored the most goals. The scoreboard shows the names of the players and the minutes they scored. Morgan scored at 12', 53', 74', 81', and 87', totaling 5 goals. Lavelle scored at 20' and 56', totaling 2 goals. Horan scored at 32', totaling 1 goal. Mewis scored at 50' and 54', totaling 2 goals. Rapinoe scored at 79', totaling 1 goal. Therefore, Morgan scored the most goals in the game.
<Answer>C. MORGAN

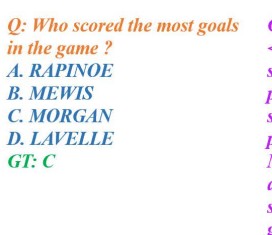

Q: How many kilometers are they from the final finish line?
A. 3.095km
B. 3.155km
C. 2.730km
D. 2.545km
GT: A

GPT-4o:
<Reason>The image shows a distance marker indicating "39.1KM." A standard marathon is 42.195 kilometers long. To find out how many kilometers are left to the finish line, I subtract the distance already covered from the total marathon distance: 42.195 km - 39.1 km = 3.095 km.
<Answer>A

Figure 29: Example of GPT-4o Sport Scenario Correct Answers.

**Transport:**

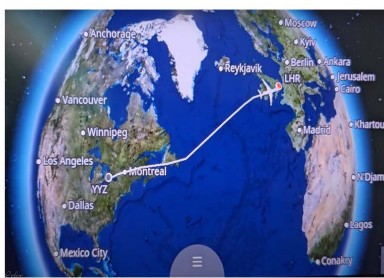

*Q: What ocean does this flight mainly cross over?*
*A. Indian Ocean*
*B. Arctic Ocean*
*C. Atlantic Ocean*
*D. Pacific Ocean*
*GT: C*

*GPT-4o:*
*<Reason>The flight path shown in the image is from Toronto (YYZ) to London Heathrow (LHR). The route crosses over the ocean between North America and Europe. The ocean located between these two continents is the Atlantic Ocean.*
*<Answer>C. Atlantic Ocean*

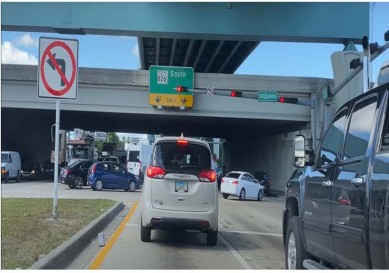

*Q: Which highway will we enter if we continue on this road?*
*A. SR 826 South*
*B. SR 826 North*
*C. SR 816 North*
*D. SR 816 South*
*GT: A*

*GPT-4o:*
*<Reason>The image shows a road sign indicating that the lane we are in leads to SR 826 South. The sign is directly above the lane. This indicates that continuing on this road will lead us to SR 826 South.*
*<Answer>A. SR 826 South*

Figure 30: Example of GPT-4o Trans Scenario Correct Answers.

# E  BAD CASE

In this section, we will conduct a case study of the types of errors that different models make in each dimension of MDI-Benchmark. We classify errors into three categories: information extraction errors, lack of knowledge errors, and reasoning errors. Errors are highlighted in red.

**Information Extraction Error.** As shown in Figure 31. It occurs most frequently. This is because the visual encoder of LMMs often fails to correctly capture the content information in the images, leading to incorrect answers.

***Information Extraction Error***

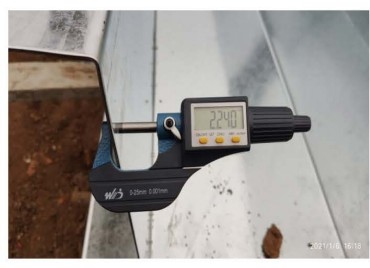

*Q: This micrometer reads*
______ *mm*
*A. 2.24*
*B. 22.4*
*C. 224*
*D. 2.42*
*GT: A*

*GPT-4o:*
*<Reason>The digital display on the micrometer shows a reading of 22.407 mm. The options provided are A. 2.24 mm, B. 22.4 mm, C. 224 mm, and D. 2.42 mm. The closest match to the reading on the display is option B. 22.4 mm.*
*<Answer>B. 22.4*

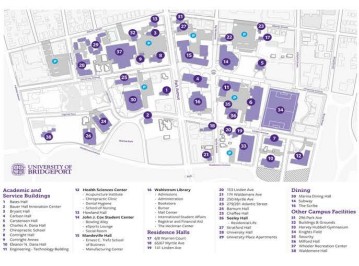

*Q: According to the map of the university, how many parking lots are there?*
*A. 10*
*B. 9*
*C. 8*
*D. 7*
*GT: D*

*Gemini-1.5-pro : <Reason> I will count the number of parking lots marked with a "P" on the map of the University of Bridgeport. I will identify and count each one to determine the total number of parking lots. After counting, I see there are 8 parking lots.*
*<Answer>C. 8*

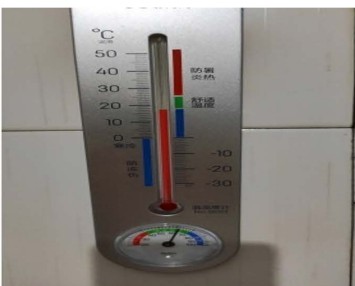

*Q: What's the room temperature now?*
*A. -10℃*
*B. 17℃*
*C. 25℃*
*D. 33℃*
*GT: B*

*Phi-3-Vision:*
*<Reason> The thermometer in the image shows a red liquid level indicating the current temperature. The red liquid level is aligned with the 25℃ mark on the thermometer scale.*
*<Answer>C. 25℃*

Figure 31: Example of Information Extraction Error.

**Knowledge Deficiency Error.** As shown in Figure 32. Because LMMs lack the ability to associate and search for relevant knowledge within certain contexts. For example, when presented with an image of a past sports event, the model fails to provide the final score.

*Knowledge Deficiency Error*

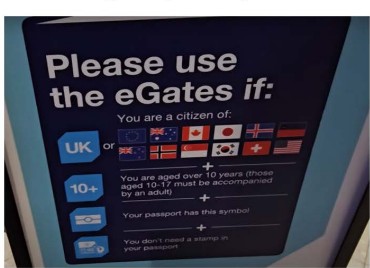

*Q: How many countries can citizens use this channel?*
*A. 24*
*B. 13*
*C. 12*
*D. 39*
*GT: C*

*Qwen-VL-Plus:*
*<Reason>I will count the number of flags shown on the sign to determine how many countries' citizens can use the eGates. The flags represent the countries whose citizens are eligible. There are 13 flags in total.*
*<Answer>B. 13*

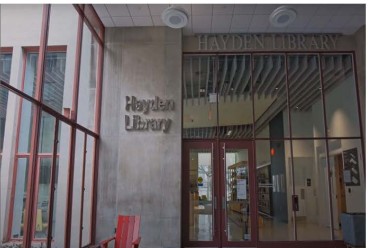

*Q: I'm visiting the Massachusetts Institute of Technology and want to know the opening hours of the facility in front of me?*
*A. 7am - 9pm*
*B. 7am - 7pm*
*C. 9am - 9pm*
*D. 24/7*
*GT: C*

*LLaVA-NeXT-110B:*
*<Reason>To determine the opening hours, I need to consider typical library hours and possibly look for any visible signs or information in the image. Libraries often have extended hours but are not typically open 24/7. the most likely opening hours for a university library would be either 7am - 9pm*
*<Answer>A. 7am - 9pm*

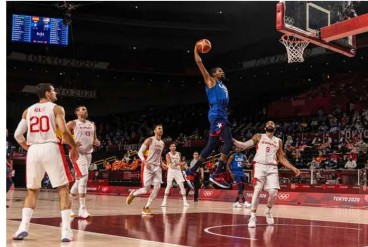

*Q: What was the final score of the game?*
*A. 81-95*
*B. 101-103*
*C. 94-102*
*D. 76-83*
*GT: A*

*MiniCPM-V2.5:*
*<Reason>The image shows a basketball game during the Tokyo 2020 Olympics. The scoreboard in the background displays the score, which appears to be 94-102. This matches one of the provided options.*
*<Answer>C. 94-102*

Figure 32: Example of Knowledge Deficiency Error.

**Reasoning Error.** As shown in Figure 33. LMMs correctly extract relevant visual information from the image but make mistakes during the reasoning process, leading to incorrect answers.

*Reasoning Error*

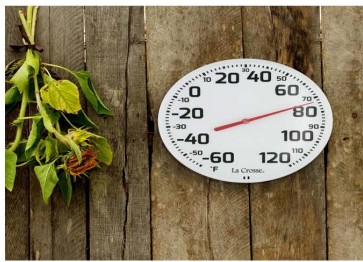

*Q: The room temperature right now is _____ degrees Fahrenheit?*
*A. 72*
*B. 74*
*C. 76*
*D. 78*
*GT: A*

*LLaVA-NeXT-72B :*
*<Reason>The image shows a thermometer with the needle pointing to the first cell at 70 degrees Fahrenheit, between 70 and 80 degrees Fahrenheit, and each cell is 4 degrees Fahrenheit, so the final result is 74 degrees Fahrenheit.*
*<Answer>B*

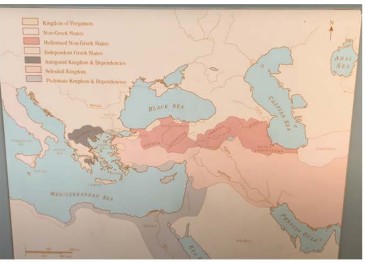

*Q: The map shows _____ different dynasties and states?*
*A.6 B.7 C.8 D.9*
*A. 6*
*B. 7*
*C. 8*
*D. 9*
*GT: B*

*CogVLM-chat:*
*<Reason>I will count the different colors and labels on the map that represent different dynasties and states. Kingdom of Pergamon, Non-Greek States, Hellenised Non-Greek States, Independent Greek States, Antigonid Kingdom & Dependencies, Seleukid Kingdom, and Ptolemaic Kingdom & Dependencies. Thus, the legend indicates six categories*
*<Answer>A*

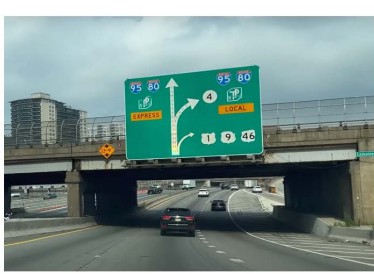

*Q: I want to enter the city on Route 4, which exit should I take?*
*A. this one*
*B. next one*
*C. uncertain*
*GT: B*

*mPLUG-Owl2-7B:*
*<Reason>The sign indicates that Route 4 can be accessed by taking the exit on the right, which is marked as "LOCAL." This suggests that the current exit is the correct one to take to enter the city on Route 4.*
*<Answer>: A. this one*

Figure 33: Example of Reasoning Error.

