# OpenReview forum: "Multi-Dimensional Insights: Benchmarking Real-World Personalization in Large Multimodal Models"
_ICLR.cc/2025/Conference — Submitted to ICLR 2025_

### Official Review · Reviewer_xJ8b · 2024-11-03

**Soundness:** 3
**Presentation:** 3
**Contribution:** 2
**Rating:** 5
**Confidence:** 4

**Summary:**

The paper presents the Multi-Dimensional Insights (MDI) Benchmark, developed to address the limitations in existing evaluations of Large Multimodal Models (LMMs) by assessing real-world personalization. This benchmark includes over 500 images across six key life scenarios, each accompanied by a set of questions that vary in complexity and are tailored to different age groups: young, middle-aged, and older adults. The MDI-Benchmark evaluates both basic perceptual capabilities (like object recognition) and more complex skills, such as reasoning and knowledge application. The study finds that while advanced models like GPT-4o demonstrate high performance, there is substantial room for improvement, especially in adapting to diverse age-related needs. The paper's contributions include providing a personalized benchmark and revealing insights into LMMs' real-world applicability, which could guide the development of more personalized AI systems.

**Strengths:**

The MDI-Benchmark evaluates multimodal models’ personalization across different age groups and life scenarios, filling a gap in current model evaluation. It sets a precedent for multi-dimensional, population-specific model testing. The paper is well-structured, with clear descriptions of evaluation dimensions (scenario, complexity, age group), and the tables and charts are easy to understand. The appendix provides examples, making it easier for readers to grasp the experimental content.

**Weaknesses:**

•	1. Limited Data Coverage and Sample Representativeness:
Benchmark should be a tool to comprehensively evaluate a certain capability of the model. Even if some shortcomings are mentioned in the limitation of the article, the MDI-Benchmark includes only around 500 images and 1,200 questions, which is relatively limited given the paper’s aim to address real-world personalization across diverse scenarios. This sample size may not capture the full range of complexities and variability in real-world interactions. To enhance representativeness, the paper could expand the dataset by incorporating more images and questions across a broader range of subdomains and scenario-specific contexts. This would make the benchmark more robust in testing model adaptability across nuanced human needs.


•	2. Poor scalability. There is no way to dynamically add some new real-life scenarios. The scenarios are pre-defined. Compared with the existing datasets for evaluating the basic capabilities and reasoning capabilities of LMM, the scale of dataset is too small. If we can analyze how the proposed dataset can improve the performance of the model in real-life scenarios and improve the personalized performance of the model.


•	3. Over-reliance on Multiple-Choice Questions:
Although multiple-choice questions offer a straightforward evaluation metric, they limit the depth of assessment, particularly in evaluating complex reasoning, creativity, and nuanced language understanding in multimodal models. To better assess these dimensions, the paper could diversify its evaluation methods, such as open-ended questions or tasks that require models to generate explanations or make decisions based on scenario analysis. This would provide richer insights into models' real-world reasoning abilities and capacity for personalized responses.


•	4. Limited Scope of Personalization Dimensions:
The benchmark primarily focuses on age as a dimension for evaluating personalization, which may not fully capture the variety of personalized needs in real-world applications. While age is a relevant factor, additional dimensions like cultural background, professional domain, and personal preferences could significantly enhance the depth of analysis. Adding these dimensions, even in a limited capacity, would align better with the goal of creating models that are adaptable to diverse user bases and provide more actionable insights for real-world AI personalization.


•	5. Sample Size Limitation:
Although the study’s sample of 2,500 survey responses offers a substantial base, it may not be sufficient to capture the full diversity and complexity needed for a benchmark intended to represent “real-world personalization.” Social science research often requires larger sample sizes when generalizing findings across highly variable populations, particularly for studies spanning multiple age groups, professions, and cultural backgrounds. Increasing the sample size, especially within each subgroup (age, gender, occupation), could help ensure that the collected data more accurately represents the broader population, strengthening the benchmark’s reliability.


•	6.Potential for Survey Response Bias:
Relying on self-reported survey data introduces the risk of response bias, as participants may interpret or prioritize certain scenarios based on personal experience, cultural background, or social desirability. This can lead to skewed data that may not accurately reflect the actual needs and preferences of the population. To mitigate this bias, the paper could supplement survey data with observational or experimental data, allowing for a more objective assessment of real-world needs. Additionally, employing stratified sampling or weighting techniques to adjust for potential demographic imbalances in survey responses could reduce bias and improve data representativeness.


•	7. Questionnaire Design Limitations:
The process of summarizing responses into selected sub-domains may inadvertently overlook less common, but still important, scenarios. Standardizing questions to reduce interpretation variance and piloting the questionnaire in diverse groups could improve question reliability. Furthermore, expanding the scope of questions to address a wider range of sub-domains would enhance the comprehensiveness of the MDI-Benchmark, ensuring a better reflection of real-world complexities in model testing.

**Questions:**

Please see weaknesses for details.

---

> ### Author Response · Authors · 2024-11-23
> **Response 1/2**
>
> Dear reviewer xJ8b,
>
> We sincerely appreciate the time and effort you have dedicated to reading our manuscript with such care and the insightful comments you have provided. Your thorough examination has been instrumental in helping us identify areas for improvement, and we are grateful for the opportunity to address your concerns and enhance the quality of our work. Below is our response to your comments.
>
> ## W1: Limited Data Coverage and Sample Representativeness
> We understand your concerns regarding data coverage in benchmark datasets, as it is crucial for evaluating model performance. However, we also aim to use high-quality data that accurately reflects the model's effectiveness. In this paper, as outlined in Section 3.2, line 216, we employed rigorous criteria for sampling from the population and ultimately collected data for the MDI-Benchmark. Our goal was to ensure the representativeness and stability of our samples by including 514 images and 1,298 questions across various scenarios, age groups, and ability dimensions.
>
> We recognize that increasing the sample size requires significant time and human resources. We are committed to continuing this deep exploration in future research, expanding both the subfields and the image-question pairs. This will help ensure that our evaluations are more aligned with real-world scenarios and possess greater stability.
>
> ## W2: Poor scalability
> Existing high-quality datasets often include images sourced from previous datasets, whereas the MDI-Benchmark mentioned in this paper derives its images from real individuals across different age groups, undergoing multiple rounds of high-quality screening. Additionally, the various domain scenarios we have set are not immutable; as noted in lines 192-193 of the paper, the real-life scenarios included in the MDI-Benchmark were selected based on survey results reflecting common situations in human society. Clearly, these scenarios cannot encompass all aspects of human life. We are very open to expanding the dataset in future research to include more images and questions, allowing for a broader range of domains and scenarios to better evaluate LMMs.
>
> Based on the results obtained from analyzing these different scenarios using the MDI-Benchmark, as mentioned in line 354, we found that there is still room for improvement in most LMMs within the sports domain. Furthermore, as noted in line 486, there is potential for growth in aligning with the values of middle-aged individuals. This insight encourages us to focus more on these aspects during the training and alignment of LMMs, aiming to enhance their personalization capabilities further.
>
> ## W3: Over-reliance on Multiple-Choice Questions
> In this paper, we chose multiple-choice questions due to their directness and convenience for evaluation. Additionally, multiple-choice questions have been widely used in existing multimodal assessment standards [1, 2, 3], providing a clear quantitative metric for determining model performance.
>
> Regarding the open-ended question evaluation method you proposed, while it can delve deeper into a model's language understanding and creative thinking abilities, it also presents new challenges. This is because the answers to open-ended questions may be influenced by various factors. If automated evaluation methods, such as those using GPT, are employed, they could introduce biases that affect the consistency and reliability of the evaluation results.
>
> Therefore, considering the operational feasibility and standardization of the assessment, we ultimately opted for multiple-choice questions. We greatly appreciate your valuable feedback, and in the future, we plan to introduce a more diverse range of evaluation methods into our assessments. We will also explore how to reduce bias through standardized scoring criteria and cross-validation among multiple evaluators, aiming for a more comprehensive evaluation of multimodal models' performance in aligning with human needs.
>
> ## references
> [1]Ying K, Meng F, Wang J, et al. Mmt-bench: A comprehensive multimodal benchmark for evaluating large vision-language models towards multitask agi[J]. arXiv preprint arXiv:2404.16006, 2024.
>
> [2]Yue X, Ni Y, Zhang K, et al. Mmmu: A massive multi-discipline multimodal understanding and reasoning benchmark for expert agi[C]//Proceedings of the IEEE/CVF Conference on Computer Vision and Pattern Recognition. 2024: 9556-9567.
>
> [3]Li B, Wang R, Wang G, et al. Seed-bench: Benchmarking multimodal llms with generative comprehension[J]. arXiv preprint arXiv:2307.16125, 2023.

---

> > ### Author Response · Authors · 2024-11-23
> > **Response 2/2**
> >
> > ## W4: Limited Scope of Personalization Dimensions
> > We understand your concerns that these dimensions may not fully support an in-depth evaluation of large multimodal models (LMMs). Your suggestion to incorporate these different dimensions into the evaluation framework is indeed insightful, as it could enhance our understanding of how LMMs align with the values of real humans.
> >
> > However, given that each social member's role is constantly shifting and is more challenging to quantify compared to age, this could lead to data bias during the construction of the evaluation dataset. Therefore, we have not integrated these perspectives with age dimensions in this paper. Nonetheless, we will consider these aspects in our future work and explore how they may influence the performance of LMMs.
> >
> > We aim to expand our benchmarking by automating the construction of large-scale, high-quality datasets that align with the preferences of different populations, thereby incorporating a broader range of personalization dimensions.
> >
> > ## W5: Sample Size Limitation
> > We appreciate your concern regarding the sample size in our study. Although our sample size may not match that of other large-scale evaluation benchmarks, it was collected using carefully designed and meticulously executed sampling methods. This strategy ensures the quality and representativeness of the sample, allowing each entry to provide valuable insights into the evaluation of LMMs' capabilities.
> >
> > As mentioned in line 218, our sample collection process particularly focused on covering multiple age groups to ensure that the benchmark testing represents a broad range of demographic characteristics. We acknowledge that increasing the sample size could further enhance the generalizability and reliability of our research. Therefore, we plan to expand our sample size in future studies to cover a wider array of subgroups and further validate the robustness of our benchmark testing.
> >
> > ## W6: Potential for Survey Response Bias
> > To prevent potential data bias in the survey process, we employed neutral language in our questionnaire design to avoid biases that might arise from leading questions. We also assured respondents of their anonymity to encourage candid responses. Additionally, we ensured a diverse range of professions and social backgrounds among our survey participants to minimize scenario selection bias stemming from personal experiences. These selections from various demographics provide a more accurate representation of real-life scenarios, which were then distilled into the 18 categories proposed in our MDI-Benchmark. We believe that by sampling a broad spectrum of the population, we have effectively mitigated biases in scenario selection and ensured a high level of data representativeness.
> >
> > ## W7: Questionnaire Design Limitations
> > Your concerns are very valid, and we are indeed addressing them. In this paper, as mentioned in line 222, during the questionnaire design phase, we provided standardized questions for respondents to choose from, while also including expandable options that allowed respondents to input their own scenarios of interest. We conducted multiple rounds of small-scale questionnaire trials to refine the questionnaire settings, ultimately ensuring the quality and representativeness of the survey.
> >
> > During the survey phase, we collected 2,374 valid responses and analyzed the top 18 frequently mentioned sub-scenarios, which included both user selections and custom inputs. These were ultimately categorized into six major life scenarios.
> >
> > Your suggestions are also very constructive. In our future research, we will continually expand our research scenarios to construct a more complex evaluation set that closely aligns with real-world situations. However, the challenge lies in the fact that as the sample size increases, more time and resources are needed for designing, distributing, collecting, and analyzing the data. Therefore, given our limited resources, we prioritized ensuring data quality through meticulously designed questionnaires and rigorous data collection processes. We hope this addresses your concerns.

---

### Official Review · Reviewer_MwJu · 2024-11-05

**Soundness:** 3
**Presentation:** 3
**Contribution:** 2
**Rating:** 5
**Confidence:** 3

**Summary:**

This paper proposes the MDI-Benchmark, a tool designed to evaluate the capability of Large Multimodal Models (LMMs) in addressing human demands of personalization within multi-dimensional scenarios. The proposed benchmark comprises over 500 images and 1.2k corresponding requirements, encompassing six major aspects of human life, with a special treatment on question sampling based on three age groups, as well as question difficulty. With the MDI-Benchmark, 14 existing LMMs are evaluated, revealing their performance preferences in different scenarios.

**Strengths:**

1. It is good to evaluate the LLMs with multiple dimensions, as one single dimension is insufficient to understand the insides of the models.
2. The complexity dimension and age dimension, considered in this work, are of course dimensions that should be considered.

**Weaknesses:**

While it is interesting to see that the capacities are different among different age groups, the observations drawn from the results are not surprising to me, especially for the complexity dimension, since the LMMs have no reason to perform better in case of problems with higher complexity level. Even for the age group dimension, I will also guess the models should generally work better for young people, as they are the main generator of the data for LMMs model training.

I am not convinced that the bi-level complexity and tri-age group dimensions used in this work are sufficient to support insightful evaluations of LMMs. Actually, it is a good topic to study what’s the complete dimension set of a benchmark for LMMs evaluation. Besides age groups, there are many other dimensions to explore, e.g., the gender, the occupations, the races, the nationalities, the religion, the personality…. Have those dimensions considered in the literature? It is better the authors can discuss why they chose only age as the primary demographic dimension among the above-mentioned options. The authors can also include a literature review on personalization dimensions that have been explored in the existing LMMs evaluation benchmarks.

**Questions:**

See the weaknesses part please.

---

> ### Author Response · Authors · 2024-11-23
> **Response 1/1**
>
> Dear Reviewer MwJu，
>
> We have carefully considered your comments and have detailed our revisions and responses below.
>
> ## W1: LMMs Capability Verification
>
> Thank you for your insights on our research; we wholeheartedly agree. Previous studies have explored the value biases of LLMs across different age groups [1], confirming that LLMs align more effectively with the values of younger individuals. Similarly, it is crucial to investigate the alignment of values in the multimodal domain concerning age. We hope that LMMs can effectively adapt to the needs of all age groups, thereby enhancing our understanding of the interactions between different age users and LMMs.
>
> Thus, constructing an age-sensitive dataset to examine biases in LMMs when addressing the needs of various age groups is an urgent issue that needs to be addressed. Ultimately, as shown in Table 4 of our paper, the evaluation results from the MDI-Benchmark corroborate both our and your hypotheses that LMMs perform better in meeting the needs of younger individuals.
>
> ## W2: Evaluation of other dimensions
> Your insights are truly enlightening, and we appreciate your considerations regarding the setup of evaluation dimensions. Factors such as race, class, gender, age, religion, and identity status of each social member significantly influence individual experiences and needs[2]. However, quantifying these dimensions poses considerable challenges, as a social member may play different roles in society, with the potential for changes over time, making it difficult to isolate and disentangle them during evaluations.
>
> In contrast, age is a more objective and quantifiable dimension. As stated in line 208, we chose age as a primary dimension because it is a widely recognized and easily measurable indicator closely related to an individual's cognitive abilities, life experiences, and needs. Additionally, age data is relatively easy to obtain and is less sensitive during the dataset construction process. We have also made efforts to minimize the influence of other dimensions while collecting and formulating our questions.
>
> Currently, there are excellent works in the LMM field attempting to study social groups and personalization [3, 4, 5]. We aspire to expand this to a multidimensional perspective, constructing a larger and more comprehensive dataset that aligns with the preferences of diverse populations. This will enhance our benchmark testing and allow us to explore how these factors influence LMMs' performance, propelling advancements in personalized research.
>
> ## references
> [1]Liu S, Maturi T, Yi B, et al. The Generation Gap: Exploring Age Bias in the Value Systems of Large Language Models[C]//Proceedings of the 2024 Conference on Empirical Methods in Natural Language Processing. 2024: 19617-19634.
>
> [2]Harris A, Bartlow S. Intersectionality: Race, gender, sexuality, and class[J]. Handbook of the sociology of sexualities, 2015: 261-271.
>
> [3]Ge T, Chan X, Wang X, et al. Scaling synthetic data creation with 1,000,000,000 personas[J]. arXiv preprint arXiv:2406.20094, 2024.
>
> [4]Woźniak S, Koptyra B, Janz A, et al. Personalized large language models[J]. arXiv preprint arXiv:2402.09269, 2024.
>
> [5]Zhuang Y, Sun H, Yu Y, et al. Hydra: Model factorization framework for black-box llm personalization[J]. arXiv preprint arXiv:2406.02888, 2024.

---

> > ### Author Response · Authors · 2024-12-02
> > **A Kind Reminder for Reading the Response**
> >
> > Dear Reviewer MwJu,
> >
> > Thank you for your insightful suggestions. We have done our best to address your concerns. Since the rebuttal period is closing very soon, could you please check the response to see whether it mitigates your concerns? We would greatly appreciate that!
> >
> > Thank you for your time and consideration, the authors.

---

### Official Review · Reviewer_nb8L · 2024-11-08

**Soundness:** 3
**Presentation:** 3
**Contribution:** 3
**Rating:** 6
**Confidence:** 3

**Summary:**

This paper proposes a new benchmark named Multi-Dimensional Insights (MDI) to support the comprehensive, objective and accurate evaluation of Large Multimodal Models (LMM). Compared to traditional benchmark, this MDI-benchmark gives each image with two types of questions and consider the factors of human age to provide the more reasonable evaluations for LMM. In addition, this paper also applies SOTA Large models to conduct extensive experiments on this benchmark so that the effectiveness of this benchmark could be demonstrated.

**Strengths:**

1.	This benchmark provides two novel perspectives (more complex questions, ages) for supporting the comprehensive, objective and accurate evaluation of Large Multimodal Models (LMM)
2.	The experiments demonstrate this benchmark could help the scores to reflect the performance of the SOTA LMM closer to the judgement in the real world.

**Weaknesses:**

1.	This paper introduces some complex scenarios, but authors do not explain the reason that these scenarios are complex enough to allow the model evaluation results to greatly reflect real world compared to other scenarios.
2.	The criteria (scores) used for evaluation seem too simple. Should other criteria be considered? In addition, in terms of scores, I guess $score_{L2}$ might be more important and should be assigned more weight, because the second tiers might be the more significant contributions in this paper.
3.	Prompt templates seemed still hand-craft, there is no automated design solutions being given.

**Questions:**

1.	Judging from the images shown by the author, the resolution and properties of these images are different. Does this mean that these factors will not affect the effectiveness of the benchmark?
2.	In this paper, we could find the importance of age factors, do you think other main factors for this similar benchmark?

**Details Of Ethics Concerns:**

I believe that this paper does not involve human subjects or raise any concerns about engineering ethics, so it does not require any examination in this regard.

---

> ### Author Response · Authors · 2024-11-23
> **Response 1/2**
>
> Dear Reviewer nb8L,
>
> We sincerely appreciate your attention to our work and your valuable feedback. Below, we provide a detailed response to your comments.
>
> ## W1: Scenarios Complexity
>
> The selection of these scenarios was not based on their complexity, as you rightly pointed out since defining the complexity between scenarios can be quite challenging. Instead, our benchmark is designed to address different scenarios with a range of complexities tailored to human requirements, allowing us to evaluate the capabilities of LMMs in responding to the needs of various age groups in real-life scenarios.
>
> As described in lines 189-196, we distilled 30 real-world scenarios from the literature [1,2,3] and conducted a rigorous survey to gather feedback from the community. From this, we identified the 18 most frequently mentioned scenarios, which we ultimately categorized into six distinct groups.
>
> Furthermore, as noted in line 218, we employed a meticulous data collection process to obtain images from real-life situations. In line 231, we ensured that these images underwent multiple rounds of cross-validation during the dataset construction process. This guarantees that all images included in our evaluation set originate from the real world and contain rich semantic information and contextual depth.
>
> Thus, the MDI-Benchmark, built using authentic images from human life scenarios, provides a reliable assessment of LMMs' ability to address the needs of different age groups in real-world contexts.
>
>
> ## W2: The criteria (scores) used for evaluation seem too simple
>
> We chose to use the score rate from multiple-choice questions as our evaluation metric because it is both intuitive and precise. Moreover, we believe that this method aligns with the approaches utilized in existing high-quality multimodal evaluation datasets, which have received widespread recognition and application in the academic community [4,5,6]. This scoring method effectively reflects the performance differences among various models.
>
> Additionally, we acknowledge your interest in the distribution of evaluation score weights. In this paper, as stated in line 202, we designate score L1 as the assessment score for the model's foundational capabilities and score L2 for its more complex abilities, such as reasoning and inference. We assert that a model must possess solid foundational skills to leverage them for complex reasoning tasks. Therefore, we assign equal weight to both scores, allowing us to evaluate the model's utilization of basic abilities alongside its comprehension of complex tasks. This approach helps maintain fairness and objectivity in our assessment, ensuring that neither aspect is unduly emphasized.
>
> We also look forward to exploring different weight distributions in future research and investigating how these variations might impact the evaluation process, further optimizing our assessment methodology.
>
>
> ## W3: Prompt Templates
> Regarding your feedback on manual prompt templates, we have explored options for automated design solutions. However, our experiments revealed that existing LMMs exhibit inconsistencies when executing instructions from automatically generated templates. On the other hand, manually designed prompt templates allow us to exert precise control over the content input into the model, thereby minimizing biases introduced by varying question formats. This precision is crucial for ensuring the accuracy and reproducibility of our experimental results.
>
> While automated prompt template construction can alleviate some workload, many successful evaluation datasets still rely on carefully crafted manual prompts [4,5,6].
>
> We are also eager to discuss the potential applications of automated prompt engineering with you. We believe there is value in exploring more automated design solutions and integrating them with manual design methods to enhance the accuracy of evaluations across different models.
>
> ## references
> [1]Tajfel H. Individuals and groups in social psychology[J]. British Journal of social and clinical psychology, 1979, 18(2): 183-190.
>
> [2]Birmingham E, Bischof W F, Kingstone A. Social attention and real-world scenes: The roles of action, competition and social content[J]. Quarterly journal of experimental psychology, 2008, 61(7): 986-998.
>
> [3]Spears R. Social influence and group identity[J]. Annual review of psychology, 2021, 72(1): 367-390.
>
> [4]Ying K, Meng F, Wang J, et al. Mmt-bench: A comprehensive multimodal benchmark for evaluating large vision-language models towards multitask agi[J]. arXiv preprint arXiv:2404.16006, 2024.
>
> [5]Yue X, Ni Y, Zhang K, et al. Mmmu: A massive multi-discipline multimodal understanding and reasoning benchmark for expert agi[C]//Proceedings of the IEEE/CVF Conference on Computer Vision and Pattern Recognition. 2024: 9556-9567.
>
> [6]Li B, Wang R, Wang G, et al. Seed-bench: Benchmarking multimodal llms with generative comprehension[J]. arXiv preprint arXiv:2307.16125, 2023.

---

> > ### Author Response · Authors · 2024-11-23
> > **Response 2/2**
> >
> > ## Q1: Image Resolution
> > Thank you for your attention to the concerns regarding the impact of image resolution on evaluation results. We understand your worries; however, we believe that different image resolutions do not undermine the validity of the benchmark.
> >
> > Firstly, we expect LMMs to effectively adapt to variations caused by differences in image resolution, as this adaptability is a crucial aspect of their generalization ability. In real-world applications, the image resolutions provided by users are not fixed, making it essential to evaluate LMMs on their capability to meet user needs and understand unseen data. Therefore, including images with varying resolutions and attributes is necessary for assessing the robustness and adaptability of the models.
> >
> > Moreover, many existing benchmarks also utilize images of different sizes, as noted in references [4, 5, 6, 7, 8, 9]. For each model being evaluated, the question-image pairs remain consistent, which means that the impact of image resolution on the evaluation process can be considered negligible.
> >
> > In summary, we believe that the differences in image resolution and attributes not only do not affect the validity of the benchmark but are also important factors in evaluating the robustness and performance of the models.
> >
> > ## Q2: Other factors besides age
> > Thank you for your question; your considerations are indeed critical. We emphasize the importance of age as a factor in our evaluation, as there are significant differences in needs and cognitive abilities across different age groups, and this factor is objective and quantifiable.
> >
> > However, we also recognize that there are other key factors that can influence evaluation results. Firstly, gender is a variable worth noting, as men and women may exhibit different behaviors and responses in certain contexts. Additionally, cultural background, social roles, and professional occupations can also affect individual needs and decision-making processes in real-life scenarios. Unfortunately, due to the variability and diversity of these dimensions, quantifying and assessing them can be challenging.
> >
> > We hope to continue this discussion with you. If we can find a way to quantify different social groups and establish corresponding models, it would represent a significant advancement. Currently, there are excellent works attempting to generate data for different groups [10]. In our future research, we plan to consider these additional factors to enrich our evaluation framework further. This will help us gain a more comprehensive understanding of how LMMs perform in the real world across diverse populations and ensure that our results have broader applicability.
> >
> > ## references
> > [4]Ying K, Meng F, Wang J, et al. Mmt-bench: A comprehensive multimodal benchmark for evaluating large vision-language models towards multitask agi[J]. arXiv preprint arXiv:2404.16006, 2024.
> >
> > [5]Yue X, Ni Y, Zhang K, et al. Mmmu: A massive multi-discipline multimodal understanding and reasoning benchmark for expert agi[C]//Proceedings of the IEEE/CVF Conference on Computer Vision and Pattern Recognition. 2024: 9556-9567.
> >
> > [6]Li B, Wang R, Wang G, et al. Seed-bench: Benchmarking multimodal llms with generative comprehension[J]. arXiv preprint arXiv:2307.16125, 2023.
> >
> > [7]Liu Y, Duan H, Zhang Y, et al. Mmbench: Is your multi-modal model an all-around player?[C]//European Conference on Computer Vision. Springer, Cham, 2025: 216-233.
> >
> > [8]Wadhawan R, Bansal H, Chang K W, et al. ConTextual: Evaluating Context-Sensitive Text-Rich Visual Reasoning in Large Multimodal Models[J]. arXiv preprint arXiv:2401.13311, 2024.
> >
> > [9]Chaoyou Fu, Peixian Chen, Yunhang Shen, Yulei Qin, Mengdan Zhang, Xu Lin, Jinrui Yang, Xiawu Zheng, Ke Li, Xing Sun, Yunsheng Wu, & Rongrong Ji. (2024). MME: A Comprehensive Evaluation Benchmark for Multimodal Large Language Models.
> >
> > [10]Ge T, Chan X, Wang X, et al. Scaling synthetic data creation with 1,000,000,000 personas[J]. arXiv preprint arXiv:2406.20094, 2024.

---

> > > ### Comment · Reviewer_nb8L · 2024-11-23
> > > **Response to authors 2/2**
> > >
> > > For Q1: Image Resolution, I feel satisfied with your responses. For Q2: Other factors besides age, I want to further know why you choose ages as the single attributions which should be discussed specially rather than other attributes. In addition, I also notice you divide three age distributions: young, middle, old. From an empirical point of view, should the number of middle samples in the benchmark become the majority? After all, the middle-aged population should be the largest in the population.
> > >
> > > Overall. My concerns have been addressed well, these new comments are just some curious issue.
> > >
> > > Best

---

> > ### Comment · Reviewer_nb8L · 2024-11-23
> > **Response to authors 1/2**
> >
> > Dear authors,
> >
> > Thanks for your response. I think you resolve my concerns greatly. I think your responses for scores and scenarios complexity are convincing. For W3: Prompt Templates, I know your main contribution is the benchmark, so you might not focus on automatic prompt templates, I just think that automatic prompt templates would be more convenient for users and would be the direction of future development of LMMs.
> >
> > Best

---

### Author Response · Authors · 2024-12-03
**Looking forward to feedback on the Responses.**

Dear Reviewers:

Hello! We have updated the responses and manuscript to your constructive and insightful comments, and we would like to kindly ask you to take a look at our responses and reevaluate our work based on our clarifications. Please let us know whether our response addresses your concerns or whether there is any further detail we can provide to help address them. We appreciate your time and consideration!

The authors.

---

### Meta-Review · Area_Chair_Q3fy · 2024-12-23

**Metareview:**

The paper introduces the MDI-Benchmark, offering novel evaluation perspectives (complexity, age) for Large Multimodal Models. It shows the potential of how the benchmark aligns model performance with real-world judgments through multi-dimensional, age-specific testing, filling existing evaluation gaps. The major concern is the method's reliance on hand-crafted prompt templates and the predictable performance across age and complexity dimensions. Additional coverage of data, diversified evaluation methods, and new personalization dimensions are suggested to accommodate real-world applicability and depth of analysis.

**Additional Comments On Reviewer Discussion:**

One reviewer is satisfied with the responses, while the other is not. There is a clear divergence in the overall rating of this paper. Since there are common concerns such as "Scope of Personalization Dimensions," the AC suggests to "reject" this paper from ICLR 2025.

---

### Decision · Program_Chairs · 2025-01-22

Reject